# GENERATIVE ADVERSARIAL REASONER: ENHANCING LLM REASONING WITH ADVERSARIAL REINFORCEMENT LEARNING

**Qihao Liu, Luoxin Ye, Wufei Ma, Yu-Cheng Chou, Alan Yuille**
Johns Hopkins University
`qliu45@jhu.edu`

## ABSTRACT

Large language models (LLMs) with explicit reasoning capabilities excel at mathematical reasoning yet still commit process errors, such as incorrect calculations, brittle logic, and superficially plausible but invalid steps. In this paper, we introduce Generative Adversarial Reasoner, an on-policy joint training framework designed to enhance reasoning by co-evolving an LLM reasoner and an LLM-based discriminator through adversarial reinforcement learning. A compute-efficient review schedule partitions each reasoning chain into logically complete slices of comparable length, and the discriminator evaluates each slice's soundness with concise, structured justifications. Learning couples complementary signals: the LLM reasoner is rewarded for logically consistent steps that yield correct answers, while the discriminator earns rewards for correctly detecting errors or distinguishing traces in the reasoning process. This produces dense, well-calibrated, on-policy step-level rewards that supplement sparse exact-match signals, improving credit assignment, increasing sample efficiency, and enhancing overall reasoning quality of LLMs. Across various mathematical benchmarks, the method delivers consistent gains over strong baselines with standard RL post-training. Specifically, on AIME24, we improve DeepSeek-R1-Distill-Qwen-7B from 54.0 to 61.3 (+7.3) and DeepSeek-R1-Distill-Llama-8B from 43.7 to 53.7 (+10.0). The modular discriminator also enables flexible reward shaping for objectives such as teacher distillation, preference alignment, and mathematical proof-based reasoning.

## 1 INTRODUCTION

Large language models (LLMs) have demonstrated remarkable mathematical reasoning abilities, often achieving expert-level performance across diverse benchmarks (Achiam et al., 2023; Dubey et al., 2024; Shao et al., 2024; DeepSeek-AI, 2025). However, despite extensive training on large-scale datasets with sophisticated paradigms, these models still suffer from errors in reasoning, such as incorrect calculations, flawed logic, superficially plausible but invalid arguments, and repetitive or incoherent reasoning steps. To tackle these challenges, researchers have explored approaches such as model debate collaboration, in which models debate against each other (Du et al., 2023; Liang et al., 2023) or with themselves (Kuba et al., 2025; Liu et al., 2025a), and Process Reward Models (Lightman et al., 2023; Wang et al., 2023), which aim to identify and mitigate process errors throughout the reasoning process. These methods provide finer-grained supervision and contribute to more robust and reliable LLM performance.

Among existing approaches, Process Reward Models (PRMs) have shown strong results on complex reasoning tasks, largely because they leverage detailed step-level annotations. However, PRMs face challenges related to annotation costs and data quality (Lightman et al., 2023), as fine-grained labels are expensive and prone to subjective error, and are sometimes susceptible to over- or under-reward issues (Wen et al., 2024; Lv et al., 2025). Alternatively, prompt-based methods employ LLMs as critics for stepwise judgments at a lower cost (Zhang et al., 2024; Gao et al., 2024; Xia et al., 2025). However, their judgments can sometimes be noisy, inconsistent, and less discriminative.

To bridge this gap, we retain a stepwise critic (referred to as the discriminator) but enable it to co-evolve with the LLM reasoner through joint training, generating effective step-level signals with

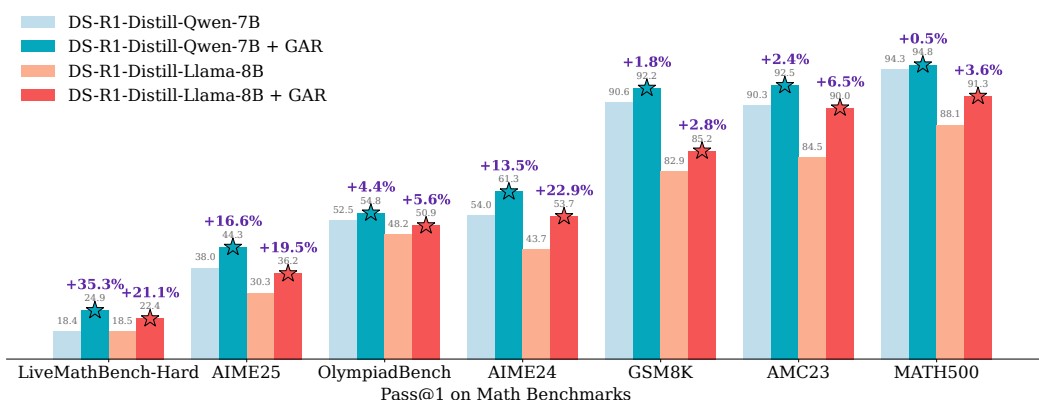

Figure 1: **Pass@1 accuracy on seven mathematical reasoning benchmarks**. Our Generative Adversarial Reasoner (GAR) consistently improves over strong baselines across both Deepseek-R1-Distill-Qwen-7B and Deepseek-R1-Distill-Llama-8B. GAR achieves gains of +22.9% on AIME24 and +19.5% on AIME25 for the Llama backbone, as well as +35.3% on LiveMathBench-Hard for Qwen. These results demonstrate the robustness and generality of GAR in enhancing reasoning performance across diverse mathematical tasks (Tab. 1).

lower annotation costs and increased robustness to label noise and reward mis-specification. Concretely, we optimize the LLM reasoner and an LLM-based discriminator together: the discriminator judges the logical soundness of each intermediate reasoning step and explains its judgment, while the reasoner learns to produce steps the discriminator consistently endorses for valid logic. This co-adaptation dynamically aligns the reward signal with the model's evolving capabilities, reduces reliance on costly fine-grained annotations, and mitigates miscalibration by continually recalibrating the discriminator to the reasoner's step distribution. As a result, we obtain better-calibrated, on-policy stepwise judgments, thereby enhancing the reasoning capabilities (Fig. 1).

However, jointly training the LLM reasoner and its stepwise discriminator introduces several challenges. First, stepwise analysis over long, intricate reasoning chains during training increases computational cost and system complexity. Second, ensuring the discriminator's judgments are rigorous and interpretable requires explicit scoring rubrics and structured rationales. Third, co-evolution can invite reward hacking: the discriminator may drift toward overly positive judgments, while the LLM may learn to produce reasoning steps that look plausible but are semantically shallow.

To address the challenges, we propose **Generative Adversarial Reasoner (GAR)**, which incorporates a compute-efficient review schedule and an adversarial co-training framework. Specifically, we partition each reasoning chain into logically complete slices of comparable length. The discriminator evaluates each slice for logical soundness and generates a concise, structured rationale, providing localized and verifiable feedback on specific reasoning errors. For learning, we jointly update the LLM reasoner and the discriminator in an adversarial reinforcement learning scheme inspired by GANs (Goodfellow et al., 2014). The reasoner is rewarded for logically consistent steps that lead to correct final answers. The discriminator receives two complementary rewards: an alignment reward for correctly detecting errors in the reasoning, and a discriminative reward for distinguishing the reasoner's trajectories from reference rationales. Together, these signals improve sample efficiency, and deliver calibrated stepwise supervision under a controlled compute budget.

Compared with previous methods, our model offers three key advantages: (i) slice-level evaluation that simplifies the discriminator's task and yields localized, interpretable justifications; (ii) on-policy joint updates that keep rewards aligned with the model's current behavior and support continued improvement, with the discriminator evolving to detect subtler errors as the LLM grows stronger; and (iii) dense step-level rewards that augment sparse exact match grading with continuous signals based on the fraction of correct steps, improving credit assignment and sample efficiency.

Our experiments show that, even compared to strong baselines (DeepSeek-AI, 2025), we achieve further improvements across various mathematical reasoning benchmarks, delivering significant gains over standard RL post-training approaches. For instance, on AIME24, we boost DeepSeek-R1-Distill-Qwen-7B from 54.0 to 61.3 (+7.3) and DeepSeek-R1-Distill-Llama-8B from 43.7 to 53.7 (+10.0), with comparable training time. These results highlight the consistent and substantial enhancements in LLM reasoning with our GAR, achieved within a comparable compute budget.

## 2 RELATED WORK

**Reinforcement Learning with Process Feedback.** Process-level supervision improves reasoning by evaluating intermediate steps rather than outcomes (Lightman et al., 2023; Uesato et al., 2022; Ouyang et al., 2022). Beyond costly human PRMs, automatic judges lower labeling cost but introduce noise; methods include Monte-Carlo from final answers, LLM-as-judge, and formulating reward modeling as next-token prediction to improve stability (Zhang et al., 2025; 2024; Gao et al., 2024; Xia et al., 2025; Xi et al., 2024). Math verifiers complement PRMs and favor concise, structured rationales (Cobbe et al., 2021). RL methods such as DeepSeek-R1 (DeepSeek-AI, 2025) and FastCuRL (Song et al., 2025) encourage behaviors like self-reflection and verification but still rely on static rewards or fixed critics that drift with the policy. Our approach jointly trains the reasoner and a discriminator under an on-policy scheme, yielding dense slice-level rewards with a compute-efficient review schedule and alignment regularization to curb reward hacking, addressing PRM mis-specification and noisy signals without sacrificing efficiency (Wen et al., 2024; Lv et al., 2025).

**Self-Play, Multi-Agent, and Game-Theoretic Training.** Self-play is a unifying mechanism: adversarial dynamics induce curricula, enabling iterative improvement from self-generated data with evolving critics (Chen et al., 2024; Yuan et al., 2024; Chen et al., 2025), while robustness-oriented variants such as SPAG instantiate these dynamics via adversarial word games (Cheng et al., 2024). Extending to the *zero-data* regime, recent work replaces external traces with task-space exploration and language self-play (Zhao et al., 2025; Kuba et al., 2025) A complementary line recasts the adversary as agents whose discourse provides training signals, with debate-style setups improving factuality and robustness (Du et al., 2023; Liang et al., 2023) To scale these dynamics, fully online multi-agent RL systems implement distributed actor–learner frameworks with role-conditioned advantages to sustain open-ended curricula transferable to math and science, while Parallel-R1 specifically targets parallel thinking (Liu et al., 2025b; Wan et al., 2025; Liu et al., 2025a; Zheng et al., 2025). Distinct from these, we embed adversarial dynamics *inside* training by co-evolving the critic with the policy to deliver fine-grained, on-policy credit assignment, consistent with recent insights on exploration and entropy for reasoning RL (Cui et al., 2025; Cheng et al., 2025; Wang et al., 2025).

**Thinking Scaffolds and Curriculum Learning.** Reasoning can be scaffolded via distilled templates, uncertainty-aware planning, and staged curricula to stabilize early steps and diversify solution paths (Yang et al., 2024; Hu et al., 2024; Zheng et al., 2025). Building on this, game-theoretic formulations treat reasoning as an interactive, multi-round protocol; Game-of-Thought shows that such interaction improves robustness and information seeking (Kempinski et al., 2025). Complementing these macro-level curricula, verifier-guided scaffolds such as Math-Shepherd provide lightweight stepwise signals without human labels, reinforcing intermediate decisions at low supervision costs (Wang et al., 2023). Our method internalizes game theory into a discriminator and couples it with compute-efficient slice-level evaluation, providing dense, calibrated, on-policy rewards that improve mathematical reasoning and code generation within a unified framework.

## 3 GAR: GENERATIVE ADVERSARIAL REASONER

We propose a modular formulation for GAR consisting of two components: a *Reasoner*, denoted $\mathcal{M}_r$, which is a general-purpose LLM that generates reasoning processes and final answers based on user input; and a *Discriminator*, denoted $\mathcal{M}_d$, which evaluates the outputs of $\mathcal{M}_r$ slice by slice. The two models are jointly trained via reinforcement learning. We provide detailed descriptions of each model and the training procedure below.

**Reasoner.** The reasoner is implemented as an LLM that generates intermediate reasoning and final answers. In principle, any model capable of step-by-step reasoning can serve as the reasoner. In this work, to demonstrate the effectiveness of our approach, we instantiate the reasoner with several state-of-the-art, open-source reasoning models (namely, variants of the official DeepSeek-R1-Distill models (DeepSeek-AI, 2025)), and show that our framework further improves their performance.

**Discriminator.** The discriminator evaluates the quality and correctness of the reasoning process, assigning a reward signal to each generated response. In our implementation, it is instantiated as a smaller, pre-trained variant of the reasoner.

However, holistic evaluation of the entire reasoning trace with the discriminator often fails to yield reliable results. We hypothesize that lengthy and complex reasoning chains, which may span thou-

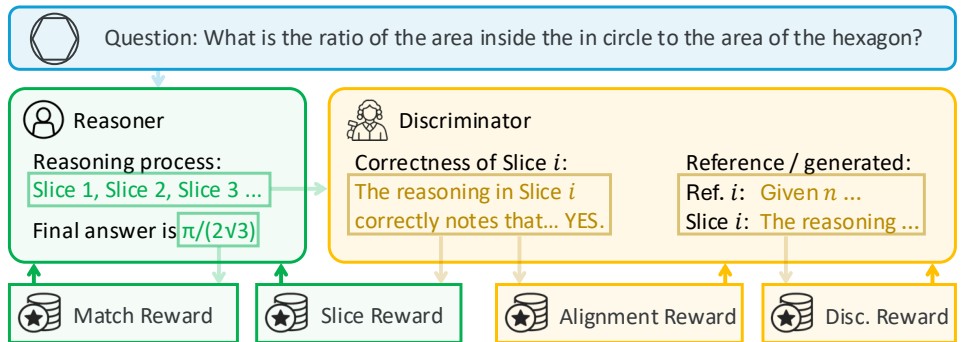

Figure 2: **GAR architecture.** GAR is a reinforcement learning framework that jointly trains an LLM reasoner and a slice-level discriminator to improve reasoning accuracy and explainability. Unlike standard RL for LLMs, which computes reward signals mainly based on final answers, GAR leverages a discriminator to provide dense, slice-level rewards that evaluate the reasoning process at each intermediate step. More importantly, the discriminator is continuously co-evolved with the LLM reasoner, ensuring the reward signal remains aligned with the model's current behavior and enabling sustained improvement and refinement of reasoning.

sands of tokens, are difficult for language models to process and evaluate faithfully, hindering precise localization of reasoning errors. To mitigate this, we partition the generated reasoning into shorter, logically coherent slices, balancing slice length and semantic completeness. Specifically, we segment the reasoning trajectory based on delimiters, then merge adjacent segments until a clear new semantic beginning is identified or a predefined token length $L = 320$ is reached. For each slice $i$, the discriminator assigns a binary slice reward $r_i^s \in \{0, 1\}$ to evaluate its reasoning quality, where $r_i^s = 1$ indicates that the slice is logically sound. The overall reward is then computed as the mean of all slice-level scores: $R^s = \frac{1}{n} \sum_{i=1}^{n} r_i^s$.

This slice-level reward mechanism offers two main advantages. *First, it improves reliability*: assessing the correctness and internal consistency of short slices is substantially easier and more accurate than evaluating a lengthy reasoning chain in its entirety. *Second, it provides a denser and more informative training signal than simple answer matching*: rather than a single binary label on the final answer, it scores the reasoning trajectory at the slice level and aggregates these scores into a fine-grained reward. Consequently, even when all final answers are wrong, the model can differentiate and reinforce better reasoning paths during RL training, improving sample efficiency and mitigating the problem of reward sparsity.

**Reward Functions.** As illustrated in Fig. 2, we jointly train the reasoner and the discriminator. For the reasoner, we use Group Relative Policy Optimization (GRPO) (Shao et al., 2024) with a reward that linearly combines (i) an exact-match term $R^m \in \{0, 1\}$, which compares the final answer to the ground truth, and (ii) the continuous reward $R^s \in [0, 1]$ from the discriminator (the mean of the slice-level scores). The overall reasoner reward is defined as $R^{\text{rea}} = \lambda_1 R^m + \lambda_2 R^s$, where $\lambda_1, \lambda_2 \geq 0$ are hyperparameters that weight the two components.

For the discriminator, we maximize two terms: a discriminator reward $R^d$ and an alignment reward $R^a$. The discriminator reward $R^d$ follows the standard GAN objective (Goodfellow et al., 2014):

$$R^d = \mathbb{E}_{x \sim p_{\text{ref}}}[\log \mathcal{M}_d(x)] + \mathbb{E}_{x \sim p_{\text{gen}}}[\log(1 - \mathcal{M}_d(x))]$$

where $\mathcal{M}_d(x)$ represents the discriminator's estimated probability that slice $x$ is real, and $p_{\text{ref}}, p_{\text{gen}}$ denote the distributions of reference reasoning slices and model-generated reasoning slices, respectively. The alignment reward $R^a$ quantifies the mean agreement between the discriminator's slice-level scores $r^s$ and the correctness of the final answer produced by the entire reasoning sequence. Under the hypothesis that correct answers are more likely to be supported by logically sound reasoning, this term encourages consistency between slice-level evaluation and answer-level correctness. The total discriminator reward is given by $R^{\text{dis}} = \lambda_3 R^d + \lambda_4 R^a$, where $\lambda_3, \lambda_4 \geq 0$ control the relative contributions. This joint training encourages the discriminator to provide calibrated, task-aligned feedback while the reasoner improves both reasoning quality and answer accuracy.

**Training Procedure.** For each batch of questions, the reasoner generates both answers and detailed reasoning steps, which we segment into multiple slices. We then mix these generated slices with an equal number of reference slices to form a balanced set and train the discriminator to distinguish

between them. The discriminator scores each slice; these scores provide the slice reward $R^s$ for the reasoner and contribute to the discriminator's own objectives ($R^d$ and $R^a$). We jointly update both models with their respective objectives and iterate, yielding improvements in reasoning quality, answer accuracy, and the discriminator's evaluation accuracy.

In addition, it is known that generating the reasoning process enhances LLMs' capabilities in handling complex tasks (Wei et al., 2022). However, this process can be computationally expensive for our task, as it requires analyzing each slice of the entire reasoning chain, potentially resulting in tens of slices per question. To improve efficiency, we modify the discriminator's workflow to (i) briefly analyze the reasoning chain, (ii) provide the evaluative judgment (slice reward $r_i^s$), and (iii) provide a concise rationale for its assessment, rather than generating a full reasoning chain before scoring. The rationale provided after the judgment is used mainly for explainability. During training, the discriminator is prompted to generate the analysis, rating (slice reward $r_i^s$), and rationale in a single response, but with the maximum generation length limited to $K = 128$ tokens to curtail the rationale and accelerate training. Notably, results in Sec. 4.3 indicate that restricting the discriminator's response at 128 tokens does not degrade performance: the final results remain comparable to those with unrestricted response lengths, while substantially accelerating training.

To further improve the evaluation accuracy, especially after switching the discriminator to an analysis–score–rationale format, we introduce a supervised fine-tuning (SFT) stage for the discriminator. In this stage, we use a pre-trained LLM to generate reasoning steps on a small subset of the training data. These reasoning steps are then evaluated by GPT-o4-mini, which provides a brief analysis, an evaluative judgment, and a concise rationale for each example. To build a balanced SFT dataset and mitigate bias, we randomly sample equal numbers of examples labeled 'yes' and 'no', ensuring both classes are equally represented. We fine-tune the discriminator on this data with early stopping, enabling it to adapt to the new format while preserving the capabilities of the original model.

In summary, training proceeds in two stages: (1) SFT of the discriminator to adapt it to the evaluation format, and (2) joint optimization of the reasoner and discriminator with GRPO. At inference time, only the LLM reasoner is used to produce answers, following the standard inference procedure.

## 4 EXPERIMENTS

This section presents a comprehensive evaluation of the mathematical reasoning capabilities of our model. Sec. 4.1 outlines the experimental setup. Sec. 4.2 demonstrates that GAR significantly improves over state-of-the-art models on mathematical benchmarks. Sec. 4.3 provides detailed analyses of GAR, and Sec. 4.4 reports ablation studies. Finally, Sec. 4.5 discusses potential applications and usage of our GAR. In the Appendix, we include additional experimental details (Sec. A and B), results on code generation (Sec. C), and ablation studies on slice-segmentation design (Sec. D).

### 4.1 EXPERIMENTAL SETUP

**Model Details.** Our implementation builds on OpenR1 (Hugging Face, 2025) and vLLM (Kwon et al., 2023), and we evaluate two backbones. For the Qwen-based setup (Qwen Team, 2024), we use DeepSeek-R1-Distill-Qwen-7B as the reasoner and DeepSeek-R1-Distill-Qwen-1.5B as the discriminator. For the Llama-based setup (Dubey et al., 2024), we use DeepSeek-R1-Distill-Llama-8B for both the reasoner and the discriminator, as no smaller Llama reasoning variant is available.

**Tasks and Benchmarks.** We evaluate GAR on mathematical reasoning tasks across seven public benchmarks: AIME 2024/2025 (MAA, a), MATH500 (Hendrycks et al., 2021), GSM8K (Cobbe et al., 2021), AMC23 (MAA, b), and LiveMathBench (Liu et al., 2024). For LiveMathBench, we evaluate its hard splits (v202505_all_en and v202412_hard_en) and report the average performance. For all evaluations, we adopt Pass@1 accuracy (averaged over 30 samples) as the metric, and fix the decoding parameters to temperature $= 0.6$, top p $= 0.95$, and max tokens $= 32K$.

**Dataset.** All experiments are conducted using the OpenR1-Math-220k dataset from the OpenR1 project. To construct instruction-tuning data for the discriminator, we randomly sample $10\%$ of the training set, partition the dataset-provided DeepSeek-R1 chains of thought into slices, and annotate each slice with binary (yes/no) judgments evaluating its soundness, along with brief rationales, using the GPT-o4-mini API. To mitigate class imbalance in these judgments, we downsample the majority class to achieve a 1:1 label ratio.

Table 1: **Pass@1 accuracy on mathematical reasoning benchmarks.** Reported scores are averaged over 30 runs per benchmark to reduce evaluation noise.

| Model | AIME24 | AIME25 | MATH500 | GSM8K | AMC23 | Olympiad Bench | LiveMath Bench-Hard |
|---|---|---|---|---|---|---|---|
| DS-R1-Distill-Qwen-7B | 54.0 | 38.0 | 94.3 | 90.6 | 90.3 | 52.5 | 18.4 |
| + GAR (Ours) | 61.3 (+7.3) | 44.3 (+6.3) | 94.8 (+0.5) | 92.2 (+1.6) | 92.5 (+2.2) | 54.8 (+2.3) | 24.9 (+6.5) |
| DS-R1-Distill-Llama-8B | 43.7 | 30.3 | 88.1 | 82.9 | 84.5 | 48.2 | 18.5 |
| + GAR (Ours) | 53.7 (+10.0) | 36.2 (+5.9) | 91.3 (+3.2) | 85.2 (+2.3) | 90.0 (+5.5) | 50.9 (+2.7) | 22.4 (+3.9) |

**Training Details.** The discriminator is first instruction-tuned with AdamW (Loshchilov & Hutter, 2017) for 500 steps (learning rate $1 \times 10^{-4}$, 100 warm-up steps, weight decay 0.0001) using a global batch size of 128 on 8 H100 GPUs. Then for adversarial reinforcement learning, we jointly optimize the LLM reasoner and discriminator for 400 steps with AdamW (initial learning rate $1 \times 10^{-6}$ with a 10% warm-up and cosine learning rate decay to $5 \times 10^{-7}$), using a global batch size of 192 on 8 H100 GPUs. Reward weights are set to $\lambda_1 = \lambda_2 = \lambda_3 = 1$ and $\lambda_4 = 0.5$.

## 4.2 ADVANCING STATE-OF-THE-ART MODELS ON MATHEMATICAL REASONING

Table 1 summarizes the Pass@1 accuracy of our method compared to strong baselines across diverse mathematical reasoning benchmarks. All results are averaged over 30 trials per benchmark, ensuring reliability by conducting three independent training runs and evaluating with 10 inference seeds per run. To ensure fair comparison, we re-evaluate all baselines under a unified evaluation protocol (Habib et al., 2023) to eliminate scripting variance. Despite starting from strong baselines, our method demonstrates consistent improvements across all benchmarks, especially on challenging datasets such as AIME24, AIME25, and LiveMathBench-Hard. For example, our approach improves the accuracy of DeepSeek-R1-Distill-Qwen-7B on AIME24 by 7.3 and on LiveMathBench-Hard by 6.5, and achieves an even larger improvement of 10.0 on AIME24 when applied to DeepSeek-R1-Distill-Llama-8B. These results highlight our method's effectiveness in addressing difficult reasoning tasks. In particular, *the discriminator model plays a crucial role by supervising reasoning traces, thereby enhancing the system's ability to solve complex questions with greater accuracy.* Beyond challenging datasets, our approach achieves notable improvements across all benchmarks, highlighting its versatility and robustness in enhancing reasoning capabilities.

## 4.3 ANALYSES AND DISCUSSIONS

This section presents a detailed analysis and discussion of the proposed method.

**Slice-Level Feedback from Our Discriminator.** Table 2 presents training-time examples of the LLM reasoning slices and our discriminator's judgments. GAR provides concise, structured assessments of each slice's soundness, yielding localized, checkable feedback. The discriminator is able to affirm correct algebraic and logical transformations, flag subtle arithmetic slips and flawed reasoning steps, and identify the exact symbols or steps responsible, while keeping rationales brief to control cost. This dense, slice-level supervision localizes errors early, improves credit assignment, and yields better learning than sparse outcome-only rewards.

**Discriminator with Improved Efficiency.** To obtain accurate and comprehensive analysis, our discriminator evaluates each slice of the full reasoning chain, which can yield tens of slices per question and increase review costs. To improve efficiency, we modify the discriminator

Table 3: **Rollout speed.**

| | AIME24 | Training Time (hours) |
|---|---|---|
| Standard RL | 56.3 | 16 |
| GAR w/ truncation | 61.3 | 19 |
| GAR w/o truncation | 60.8 | 43 |

workflow as shown in Table 2: the discriminator first gives a brief analysis, then a binary 'yes/no' verdict on the slice's soundness, and finally a concise rationale. During joint training of the LLM reasoner and the discriminator, this design yields reliable judgments without requiring a full rationale. Specifically, we cap the discriminator's output at 128 tokens, preserving the verdict and truncating any justification beyond that limit. Table 3 compares three training settings: standard RL without discriminator, discriminator with truncation, and discriminator without truncation. Applying the cap preserves accuracy while significantly improving training efficiency. It demonstrates that our analysis–score–rationale design delivers dense supervision with minimal overhead.

Table 2: **Training-time examples of LLM reasoning slices and our discriminator judgments.** We underline the discriminator's key analysis leading to its judgment. Our model provides localized, interpretable justifications for each reasoning slice. Additional examples in Appendix E.

| | LLM Reasoning Slices | Our Full Discriminator Judgments |
|---|---|---|
| 1 | Given $n = (a-b)(a+b)$, and both $a-b$ and $a+b$ are either both even or both odd. 

 If $n$ is odd, then both $a-b$ and $a+b$ are odd, and their product is $n$. So, any odd number can be expressed as such. 

 If $n$ is even, then both $a-b$ and $a+b$ must be even, so $n$ must be divisible by 4. Let's write $n = 4k$, where $k$ is a positive integer. Then, $a-b = 2m$ and $a+b = 2n$, where $m \times n = k$. Thus, $a = m+n$ and $b = n-m$. For $b$ to be positive, $n > m$, and both $m$ and $n$ must be positive integers. 

 Therefore, for $n$ to be a wise number, if $n$ is odd, it's automatically a wise number. If $n$ is even, it must be divisible by 4, and $n/4$ must be expressible as a product of two positive integers $m$ and $n$ where $n > m$. Wait, but $n/4$ just needs to have at least one pair of factors where $m < n$, which is true as long as $n/4 \geq 1$, which is always true since $n$ is positive | The reasoning correctly notes that for odd $n$, a representation always exists. For even $n$, it correctly deduces $n$ must be divisible by 4 and sets up the parameterization $a = m+n$, $b = n-m$ with $m \cdot n = k = n/4$. It requires $n > m > 0$ for $b > 0$. However, it then claims that as long as $n/4 \geq 1$, such a factor pair exists, which is not always true. **NO** 

 The reasoning is flawed in the even case. While it is true that $n$ must be divisible by 4, the conclusion that $n/4 \geq 1$ guarantees such a factorization ignores the requirement that $b = n-m$ must be positive, meaning $m < n$. However, if $n/4$ is a prime square (e.g., $n = 4$, so $n/4 = 1$), the only factor pair is $m = 1, n = 1$, which gives $b = 0$, invalid since $b$ must be a positive integer. Thus, $n = 4$ is not expressible in the required form, contradicting the claim that all even $n$ divisible by 4 with $n/4 \geq 1$ work. The reasoning fails to consider that $n/4$ must have a factor pair with distinct factors. |
| 2 | 1. $(2, 3, -4)$: Prefix sums are 2, 5, 1. All positive. Good. 
 2. $(2, -4, 3)$: Prefix sums are 2, -2, 1. The second prefix sum is negative. Not good. 
 3. $(3, 2, -4)$: Prefix sums are 3, 5, 1. All positive. Good. 
 4. $(3, -4, 2)$: Prefix sums are 3, -1, 1. The second prefix sum is negative. Not good. 
 5. $(-4, 2, 3)$: Prefix sums are -4, -2, 1. The first prefix sum is negative. Not good. 
 6. $(-4, 3, 2)$: Prefix sums are -4, -1, 1. The first prefix sum is negative. Not good. 

 So, out of 6 permutations, only 2 are good. Hmm, that's interesting. So, for $n = 3$, it seems that we can have at least 2 good permutations exist. But is this the minimum? Let's see another example to confirm. | The reasoning lists all permutations of (2, 3, -4), computes their prefix sums, and counts how many have all prefix sums positive. It finds 2 permutations are "good" and then asks if this is the minimum. **YES** 

 The thinking process enumerates all permutations of the specific numbers $(2, 3, -4)$ and correctly computes their prefix sums to determine which permutations have all prefix sums positive (only two: $[2, 3, -4]$ and $[3, 2, -4]$). Each listed prefix-sum sequence is arithmetically accurate, the classification (good / not good) follows the stated criterion, and the cautious conclusion ("at least 2") is logically sound (they do not overclaim a proven minimum). The only minor point is they stop short of proving minimality for $n = 3$, but that does not detract from the correctness of the presented analysis. |

**Selective-Entropy without Collapse.** Recent studies on RL for LLMs warn that performance gains often come with policy entropy collapse and reduced diversity, hurting exploration and calibration (Cui et al., 2025; Cheng et al., 2025; Wang et al., 2025). However, as shown in Fig. 3-(a), after RL training, our per-problem mean-entropy distribution remains comparable to the baseline model (DeepSeek-R1-Distill-Qwen-7B), with only a mild left shift and preserved spread. Moreover, contrary to the common trade-off between accuracy and entropy, we raise AIME24 accuracy from $54.0$ to $61.3$ $(+7.3)$ without a global drop in entropy. In Fig. 3-(b), both models exhibit the expected "wrong > correct" pattern. However, our wrong-case violin plot is markedly tighter with a shorter low-entropy tail, and our correct-case entropy is lower than R1's, demonstrating better calibration and fewer extreme failures. More importantly, Fig. 3-(c) (computed after removing zero-entropy tokens) reverses the ordering for correct cases: our mean entropy over non-zero-entropy tokens exceeds R1's, and is comparable between our correct and wrong groups, indicating that we suppress entropy only where the model is confident while retaining stochasticity on informative tokens.

The contrast between Fig. 3-(b) and (c) reveals a selective-entropy mechanism: our on-policy slicing with an adversarial discriminator encourages low entropy on deterministic slices (producing many zero-entropy tokens and a lower global mean) while sustaining exploration on decision-critical slices (higher non-zero entropy), thereby reducing high-entropy outliers when the model is wrong. This

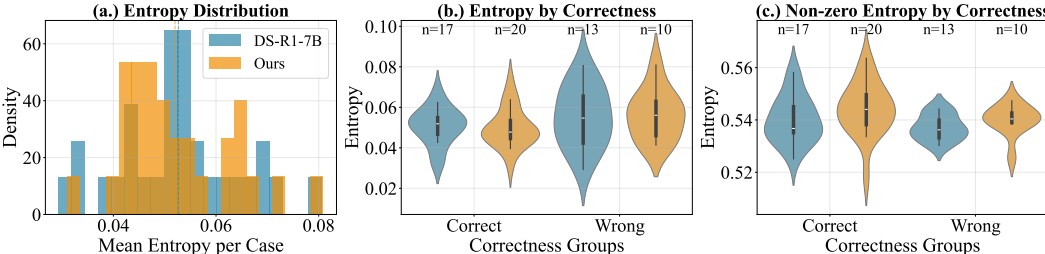

Figure 3: **Improving model without entropy collapse.** Compared to DeepSeek-R1-Distill-Qwen-7B, our method raises AIME24 accuracy (+7.3), but maintains a comparable overall mean-entropy distribution (5.20% vs. 5.27%) (a) and tightens the "wrong" distribution (b), indicating better calibration with fewer extreme-uncertainty failures. Removing zero-entropy tokens (c) flips the ordering (entropy is higher on correct cases), revealing a selective-entropy behavior (decisive on deterministic spans, exploratory on decision-critical tokens) that aligns with the AIME24 accuracy gains.

Table 4: **Ablation study.** Beginning with the baseline (DeepSeek-R1-Distill-Qwen-7B), we verify the effectiveness of each component.

| Model | Exact Match Grading $\mathcal{R}^m$ | Judger Score $\mathcal{R}^g$ | Alignment Reward $\mathcal{R}^g$ | Discriminator Reward $\mathcal{R}^g$ | AIME24 | AIME25 |
|---|---|---|---|---|---|---|
| 1 Baseline (BL) | | | | | 54.0 | 38.0 |
| 2 BL + Standard RL | ✓ | | | | 56.3 | 40.7 |
| 3 BL + Fixed Standard Critic | ✓ | ✓ | | | 56.7 | 40.4 |
| 4 BL + Fixed GAR Discriminator | ✓ | ✓ | | | 58.6 | 42.0 |
| 5 BL + Trainable GAR Discriminator | ✓ | ✓ | ✓ | | 59.4 | 42.8 |
| 6 BL + Trainable GAR Discriminator | ✓ | ✓ | | ✓ | 60.2 | 43.3 |
| 7 BL + Trainable GAR Discriminator | ✓ | ✓ | ✓ | ✓ | 61.3 | 44.3 |

pattern explains the observed accuracy gains on AIME24 and suggests a practical control signal: token- or slice-level entropy can trigger self-checks or adaptive sampling precisely where uncertainty is concentrated, improving both efficiency and reliability.

## 4.4 ABLATION STUDY

We conduct various ablations in Table 4, progressively building on the baseline DeepSeek-R1-Distill-Qwen-7B to validate each component and culminate in our final model (row 7).

**Discriminator Design Analysis.** We ablate the discriminator in Table 4 (rows 2 – 4). Row 2 is a baseline LLM fine-tuned with standard GRPO using an outcome-based exact-match reward, without any discriminator. Row 3 adds a fixed standard critic (DeepSeek-R1-Distill-Qwen-1.5B) to provide feedback. Row 4 keeps the LLM critic's capacity fixed, instruction-tunes it to our slice-level judgment format with brief rationales, and deploys it under the compute-efficient review schedule.

The table shows that row 3 improves over row 2 on AIME24, confirming the benefit of adding a discriminator. Moreover, Row 4 consistently outperforms both, indicating that the discriminator drives the gains. By reframing the discriminator's role from holistic solution grading to slice-level soundness judgments with concise rationales, we obtain more accurate and interpretable feedback. The resulting dense, slice-level rewards provide continuous learning signals compared with sparse exact-match grading, improving credit assignment and sample efficiency, and thereby significantly boosting performance across benchmarks.

**Reward for Discriminator Training.** We ablate the discriminator's reward design in Table 4 (rows 5 - 7). Both the alignment and discriminator rewards individually improve performance over the baseline. Combining them yields the best results, indicating the two signals are complementary. The alignment term sharpens the discriminator's ability to distinguish correct from incorrect reasoning, but its supervision can be noisy because it depends on the correctness of that step's generated final answer. The discriminator term stabilizes learning by steering the discriminator toward reference judgments. Together, these complementary signals yield a stronger and more reliable training signal.

**Effectiveness of Joint Training.** Finally, we evaluate joint training of the LLM reasoner and the discriminator in Table 4 (rows 4 and 7). Compared with using a fixed discriminator (row 4), joint on-policy updates (row 7) yield consistent gains by keeping rewards aligned with the reasoner's current

behavior. As the reasoner improves, the co-trained discriminator adapts to detect subtler errors and provides more informative slice-level feedback, which raises the performance ceiling and mitigates drift or overfitting to a static reward signal.

### 4.5 GAR Unlocks New Applications and Future Directions

Finally, we briefly discuss novel use cases and capabilities enabled by our method in this section.

**RL without Full Chain-of-Thought or Verifiable Final Answers.** Another advantage of our approach is that it decouples RL post-training from judgeable final answers. Standard RL-based post-training requires generating a complete chain of thought and an automatically verifiable final answer. This makes training much slower than supervised fine-tuning

Table 5: **Partial-trace evaluation without a final-answer reward yields faster training and higher accuracy than standard RL.**

|  | AIME24 | Training Time (hours) |
|---|---|---|
| Standard RL | 56.3 | 16 |
| Ours (3 slices) | 57.7 | 6 |

(which only predicts the next token) and restricts applicability to tasks with clear evaluators (e.g., requiring an execution engine for code generation and struggling with open-ended math proofs). In contrast, our model provides additional reward signals, enabling us to remove the final-answer reward and update the model solely using our discriminator's scores on intermediate reasoning. We demonstrate this advantage in Table 5. Rather than generating a complete reasoning trajectory and a final answer, we stop after three reasoning slices and have the discriminator evaluate these partial traces, providing dense early feedback without a full chain of thought. This yields substantial efficiency gains while improving accuracy: our method surpasses standard RL with significantly less training time. Moreover, because it does not rely on final-answer rewards or external executors, it naturally extends to tasks with hard-to-evaluate outputs, such as mathematical proofs.

**Distilling Reasoning Patterns with GAR.** GANs have been widely used for distilling patterns in tasks like image generation (Sauer et al., 2024). Similarly, we show that GAR enables reasoning distillation, aligning a student model's reasoning pattern with that of a teacher.

Table 6: **Distinguishability of reasoning patterns.** We report the success rate of human experts distinguishing generated reasoning from Gemini reasoning before and after GAR training.

|  | w/o GAR | w/ GAR |
|---|---|---|
| Success rate | 82.3% | 55.9% |

In a demo experiment, we use the S1K-1.1 (Muennighoff et al., 2025) dataset from the OpenR1 project, containing two reasoning trajectories, "gemini_thinking_trajectory" and "deepseek_thinking_trajectory", to train the discriminator to distinguish between the two styles. We then use this discriminator to jointly train the reasoner and discriminator within our GAR framework. Notably, the reasoner is only trained on the Math220K dataset, without exposure to the Gemini trajectory during training. In the evaluation, human experts are asked to differentiate between the generated and Gemini trajectories in a randomized side-by-side comparison. As shown in Table 6, our model significantly increases similarity to the Gemini reasoning style, reducing the distinguishability success rate from 82.3% to 55.9% (close to 50% random guess baseline), making it substantially harder for experts to tell the two apart. Experimental details are included in the Appendix A. A natural extension of GAR 's distillation ability is human preference alignment, where teacher reasoning comes from human explanations, enabling alignment with human-like reasoning. We leave empirical validation to future work.

## 5 Conclusion

In this paper, we presented an adversarial co-training framework that couples an LLM Reasoner with an LLM-based Discriminator to deliver dense, calibrated, slice-level rewards that supplement sparse exact-match grading. Our model partitions reasoning into logically complete slices, and the Discriminator provides concise and structured feedback. During training, the reasoner is rewarded for logically consistent steps that lead to correct answers, and the Discriminator is rewarded for correctly detecting reasoning errors. This design reduces annotation burden, mitigates reward misspecification and reward hacking, and improves credit assignment and sample efficiency under a controlled compute budget. Empirically, the approach yields consistent gains over strong RL baselines on mathematical tasks, including +7.3 on AIME24 for DeepSeek-R1-Distill-Qwen-7B and +10.0 for DeepSeek-R1-Distill-Llama-8B, along with better calibration and fewer extreme failures.

**Acknowledgements.** Thanks to all who supported this project and to the anonymous reviewers for their constructive comments.

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

APPENDIX

In the appendix, we provide additional information as listed below:

- Sec. A provides experimental details for reasoning distillation.
- Sec. B lists the system prompts used for the reasoner and the discriminator.
- Sec. C provides additional experimental results on coding ability.
- Sec. D presents ablation results on the slice segmentation design.
- Sec. E provides additional training-time examples of the LLM reasoning slices and our discriminator judgments.
- Sec. F discusses the limitations of our method.
- Sec. G provides the ethics statement.
- Sec. H provides the reproducibility statement.

## A EXPERIMENTAL DETAILS FOR REASONING DISTILLATION.

In the demo experiment, we use the S1K-1.1 (Muennighoff et al., 2025) dataset from the OpenR1 project, which contains two reasoning trajectories, "gemini_thinking_trajectory" and "deepseek_thinking_trajectory", to train a discriminator to distinguish between the two styles. We use the same settings as our main experiments and train the discriminator for 1,000 iterations. We then integrate this discriminator into our GAR framework and jointly train the reasoner and the discriminator under the Partial trace setting: instead of generating the full reasoning trajectory and a final answer, we stop after three reasoning slices and have the discriminator evaluate these partial traces. We train this model for 3,000 steps, and all other hyperparameters are the same as in the main experiment. The reasoner is trained only on the Math220K dataset, without exposure to the Gemini trajectory during training.

For the human preference study, we generate 200 reasoning slices, pair them side by side with the corresponding Gemini slices, and randomly shuffle the order. We then ask 10 experts from three academic institutions, including 3 undergraduate students and 7 PhD students, to first familiarize themselves with the differences between Gemini and DeepSeek-R1 reasoning styles by reviewing the S1K-1.1 dataset, and then conduct the evaluation. We compare the success rate of experts distinguishing generated reasoning from Gemini reasoning before and after GAR training, and report the results in Table 6.

## B SYSTEM PROMPTS

We provide the system prompts for the LLM reasoner and the discriminator as follows:

System prompt for the LLM reasoner:

```
You are a helpful AI Assistant that provides well-reasoned and
detailed responses. You first think about the reasoning process as
an internal monologue and then provide the user with the answer.
Respond in the following format:<think>\n...\n</think>\n<answer>
\n...\n</answer>
```

System prompt for the LLM discriminator:

```
You are an evaluator responsible for assessing whether a reasoning
/ thinking process is reasonable, rigorous, and accurate. Based on
these criteria, determine if the analysis is of high quality. First,
analyze the reasoning very briefly, then respond with '**YES**' for
high quality or '**NO**' if it is not. Finally, provide a brief but
specific explanation for your judgment. Hint: You can first summarize
the given thinking process to identify the main reasoning chain, then
analyze the reasoning chain sentence by sentence.
```

## C  Performance on Code Generation

We further evaluate our model on coding-based reasoning tasks. As shown in Table 7, our method yields substantial improvements across multiple coding benchmarks. To ensure statistical reliability, all results are averaged over **30 trials per benchmark** (three independent training runs, each evaluated with 10 inference seeds).

Experiments are conducted on the **CodeForces-CoT** dataset, where the (approximated) ground-truth reasoning traces are produced by DeepSeek-R1. The maximum rollout length during training is fixed at **800 steps**. The reward function is an **equal-weight combination** of three components:

- **CF-Code Reward**: Computes a weighted sum over public test cases, assigning 1 for pass and 0 for fail.
- **Code-Format Reward**: Assigns a reward of 1 when the model output includes a valid code-block wrapper (e.g., ``` python ```).
- **Critic Reward**: Incorporates alignment and discriminative terms, using the same structure as in the math-reasoning setting.

Table 7: Pass@1 accuracy on coding benchmarks. Similar to the results reported in the main paper, scores are averaged over 30 runs per benchmark to reduce evaluation noise. Our method significantly improves coding performance.

| Model | LiveCodeBench | HumanEval | HumanEval+ |
|---|---|---|---|
| DS-R1-Distill-Qwen-7B | 37.4 | 40.4 | 37.8 |
| **DS-R1-Distill-Qwen-7B + GAR (Ours)** | **43.6** | **42.7** | **39.3** |

## D  Ablations of Slice Segmentation Design

### D.1  Segmentation Strategy

In this section, we clarify the rationale behind adopting the proposed segmentation method based on **explicit delimiters** combined with a **token-length range**. Chain-of-thought (CoT) generated by contemporary reasoning models (*e.g.*, DeepSeek-R1, Gemini) is typically organized into coherent fragments separated by line breaks ("\n") and discourse cues such as "Wait," "Since," and "Therefore,". These natural markers provide reliable boundaries for forming semantically self-contained slices. Applying an additional token-length constraint prevents segments from being excessively short or long, while keeping computation and implementation overhead minimal.

Table 8 compares this rule-based method with two alternatives: (1) pure fixed-length token windows and (2) LLM-based semantic segmentation, where a model is prompted to automatically partition the CoT. The results show that the alternatives either disrupt coherent reasoning steps or require substantially higher training cost, without yielding performance improvements over our method.

Table 8: Ablation on different slice segmentation strategies.

| Method | AIME24 | Training Time (hours) |
|---|---|---|
| Pure fixed-length token windows | 58.7 | 19 |
| LLM-based semantic segmentation | 61.6 | 35 |
| **Ours** | 61.3 | 19 |

### D.2  Sensitivity to Slice Length Variation

We further add a sensitivity analysis over slice length. As shown in Table 9, model performance is highest and most stable when slices contain approximately **320–560 tokens**. Shorter slices often contain no explicit reasoning error, making it difficult for the discriminator to learn informative supervision signals. In contrast, very long slices typically include at least one flaw, causing most

segments to be labeled as "incorrect" and reducing label diversity, which in turn weakens the discriminator's effectiveness.

Table 9: Ablation over slice length (tokens) on AIME24.

| Slice Length | 160 | 320 | 480 | 560 | 800 | 960 | 1120 | 1440 |
|---|---|---|---|---|---|---|---|---|
| Pass@1 accuracy | 57.4 | 61.3 | 61.5 | 61.4 | 61.0 | 59.3 | 56.5 | 56.8 |

## E    SLICE-LEVEL FEEDBACK FROM OUR DISCRIMINATOR.

We provide additional examples of the LLM reasoning slices and our discriminator's judgments during joint training in Table 10 and Table 11.

## F    LIMITATIONS AND FUTURE WORK

The proposed GAR has a few remaining limitations. First, it remains challenging to balance the discriminator's reasoning depth with compute efficiency. In this work, we reformulate the think–answer format into an analysis–score–rationale format to make the discriminator more efficient while preserving its reasoning ability. Nevertheless, there is room to further improve compute-efficient reasoning. Promising directions include adaptive early-exit mechanisms, dynamic truncation of analysis conditioned on confidence, etc. Second, although we compute slice-level dense rewards, the final objective aggregates them into a single trajectory-level signal by averaging, which can dilute local credit assignment and increase variance. Better ways to leverage slice-wise information could improve training efficiency and stability.

## G    ETHICS STATEMENT

This work seeks to improve model reasoning and training efficiency, but we acknowledge risks including bias amplification, spurious correlations, reward hacking, and potential misuse for generating harmful or deceptive content. We rely on publicly available datasets (Math-220K) and adhere to their licenses, avoid collecting or exposing personal data, and assess safety and fairness where applicable, documenting observed failure modes. We will not release artifacts that materially increase misuse risk and provide usage guidelines to support responsible deployment. Downstream use should include human oversight, continuous monitoring, and domain-specific risk assessments.

## H    REPRODUCIBILITY STATEMENT

We will release code, pretrained checkpoints, and detailed scripts to reproduce all results, including data preparation, training, and evaluation. The repository will provide exact configurations (model architectures, the analysis–score–rationale discriminator format, slice-level reward computation and aggregation, loss functions, optimizers, schedules), dataset splits and preprocessing, and the evaluation harness with metrics. We document software and hardware environments (package versions, CUDA/cuDNN, GPU types), supply containerized setups (Conda/Docker), and fix random seeds; results are reported as mean over 30 runs. The implementation details are also provided in Sec. 4.1.

Table 10: **Training-time examples of LLM reasoning slices and our discriminator judgments.** We underline the discriminator's key analysis leading to its judgment. Our model provides localized, interpretable justifications for each reasoning slice.

| | LLM Reasoning Slices | Our Full Discriminator Judgments |
|---|---|---|
| 1 | Case 1: $p$ divides $2^s$. Since $2^s$ is a power of 2, $p$ must be 2. Case 2: $p$ divides $q$. Since $q$ is a prime, $p = q$. Let's explore each case. **Case 1: $p = 2$** Substituting $p = 2$ into the equation, we get: $$2^s q = 2^y - 1$$ Rearranging: $$2^y = 2^s q + 1$$ Since $2^y$ is even, the right-hand side must also be even. However, $2^s q$ is even if $s \geq 1$, so $2^s q + 1$ is odd. This is a contradiction because an even number cannot equal an odd number. Therefore, $p = 2$ is not a valid solution. | The reasoning examines the case where the prime $p$ divides $2^s$. It correctly concludes $p = 2$, substitutes into the equation $2^s q = 2^y - 1$, and argues that the left side $2^y$ is even while the right side $2^s q + 1$ is odd, leading to a contradiction. **YES** The reasoning is rigorous and accurate. It systematically considers the case $p = 2$, substitutes it into the equation, and rearranges to $2^y = 2^s q + 1$. It correctly observes that $2^y$ is even, while $2^s q + 1$ is odd (since $2^s q$ is even for $s \geq 1$ ), leading to a contradiction. The logic is clear, step-by-step, and based on fundamental parity properties, leaving no room for ambiguity. This is a high-quality analysis for this specific case. |
| 2 | Let me start with the first part. I know that a wise number can be written as $a^2 - b^2$ where $a$ and $b$ are positive integers. I can factor this expression as $(a-b)(a+b)$. So, any wise number can be expressed as the product of two integers $(a - b)$ and $(a + b)$. Since $a$ and $b$ are positive integers, $a + b$ must be greater than $a - b$, and both $a + b$ and $a - b$ must be positive integers. Moreover, both $a + b$ and $a - b$ must have the same parity because $a + b + (a - b) = 2a$ must be even, meaning their sum is even. Therefore, both $a + b$ and $a - b$ must be either both even or both odd. This implies that their product $(a - b)(a + b)$ must be either odd or divisible by 4. So, if a number is of the form $4k+2$, it cannot be expressed as the product of two integers with the same parity. For example, a number like 2 cannot be written as the difference of two squares because 2 is of the form $4k+2$ (where $k = 0$), and there are no two integers $a$ and $b$ that would satisfy $a^2 - b^2 = 2$. | The reasoning shows that $a^2 - b^2 = (a - b)(a + b)$ requires both factors to have the same parity, so the product is either odd or divisible by 4. Thus, numbers of form $4k + 2$ (like 2) cannot be wise numbers. **YES** It correctly identifies that a wise number is of the form $a^2 - b^2 = (a - b)(a + b)$ , and notes that $a + b$ and $a - b$ must be positive integers with the same parity (both even or both odd) because their sum is $2a$ (even). This leads to the key insight: the product $(a - b)(a + b)$ must be either odd or divisible by 4. Consequently, numbers of the form $4k + 2$ (which are even but not divisible by 4) cannot be expressed as such a product. The example of 2 (which is $4 \times 0 + 2$) solidifies the argument. The logic is clear, step-by-step, and based on sound number theory principles. |
| 3 | But since this is an approximation, I need to check if 2686 or 2687 is a wise number. Let me check if 2686 is a wise number. 2686 divided by 2 is 1343, which is odd. So, 2686 is 2 times an odd number, which is not a multiple of 4. Therefore, 2686 is not a wise number because it cannot be expressed as the difference of squares. Next, check 2687. Since 2687 is odd, it can be expressed as a difference of squares. So, 2687 is a wise number. Therefore, the 2015th wise number is 2687. Wait, but let me make sure. Let's verify how many wise numbers are less than or equal to 2686 and 2687. | The reasoning checks if 2686 and 2687 are wise numbers, correctly using the parity/divisibility rule. It then concludes 2687 is the 2015th wise number without verifying the count. **NO** While it correctly identifies that 2686 (even but not divisible by 4) cannot be a wise number, and that 2687 (odd) can be a wise number, it fails to verify whether 2687 is actually the 2015th wise number. The reasoning jumps to a conclusion without calculating how many wise numbers exist up to 2686 and 2687. Specifically, it does not account for the fact that not all odd numbers are necessarily wise (e.g., 1 is odd and wise, but other constraints might exist) and, more importantly, it does not compute the cumulative count of wise numbers to confirm the index. This lack of systematic counting or verification makes the reasoning incomplete and potentially incorrect. |

Table 11: **Training-time examples of LLM reasoning slices and our discriminator judgments.**
We underline the discriminator's key analysis leading to its judgment. Our model provides localized, interpretable justifications for each reasoning slice.

| | LLM Reasoning Slices | Our Full Discriminator Judgments |
|---|---|---|
| 1 | But wait, I should double-check if this is indeed the minimum. Maybe I made a mistake in applying the Lagrange multipliers. Alternatively, perhaps there's a geometric interpretation that can confirm this result. 

 Thinking geometrically, as I considered earlier, the expression represents the sum of distances from points on the coordinate axes to the point $(12, 9)$. The minimal total distance should be the straight line distance from the origin to $(12, 9)$, which is $\sqrt{12^2 + 9^2} = \sqrt{144 + 81} = \sqrt{225} = 15$. That matches the result I got using calculus, so it seems correct. 

 So, the minimal value is indeed 15. | The reasoning cross-checks a calculus result (Lagrange multipliers) with a geometric interpretation: the sum of distances from points on the axes to (12,9) is minimized by the straight-line distance from the origin to (12,9), which is 15. The results match. 
 **YES** 
 The reasoning is rigorous and accurate. It cross-validates the result obtained from calculus (Lagrange multipliers) with a geometric interpretation: the expression $\sqrt{x^2 + 144} + \sqrt{(15-x)^2 + 81}$ represents the sum of distances from a point on the x-axis to fixed points, and the minimal path is indeed a straight line from the origin to (12,9), yielding a distance of 15. This geometric insight is sound and confirms the calculus result, demonstrating thoroughness and multiple perspectives. The logic is clear and well-supported. |
| 2 | From this table, I notice a pattern. It seems that $f(n)$ alternates between 1 and 0, but it's not straightforward. Let me try to see if there's a base-3 pattern or something related to the recursive formula involving division by 3. 

 Looking at the recursive formula: 

 $$f(n) = 1 - f(n-1) \cdot f\left(\left\lfloor \frac{n}{3} \right\rfloor\right)$$ 

 This suggests that the value of $f(n)$ depends on the previous value $f(n-1)$ and the value at $\lfloor n/3 \rfloor$. 
 Let me consider the sequence more carefully. It seems that $f(n)$ is 1 except when $n \equiv 0 \mod 3$, but that's not entirely accurate because $f(3) = 0$, $f(6) = 1$, $f(9) = 1$, and $f(12)$ would depend on $f(11)$ and $f(4)$. | The reasoning observes a pattern in $f(n)$ and recalls the recursive definition $f(n) = 1 - f(n-1) \cdot f(\lfloor n/3 \rfloor)$. It tests a simple hypothesis (based on divisibility by 3) but immediately notes counterexamples (like f(6)=1, f(9)=1), showing the pattern is more complex. 
 **NO** 
 The reasoning is not rigorous or accurate. It attempts to identify a pattern (e.g., alternation or modulo 3 behavior) but quickly contradicts itself with counterexamples (e.g., $f(3) = 0$, $f(6) = 1$, $f(9) = 1$). The analysis is superficial and does not deeply engage with the recursive formula. It mentions considering the recursion but does not systematically compute values or derive properties. |
| 3 | In a geometric sequence, each term is obtained by multiplying the previous term by a constant ratio $r$. Given $a_2 = 2$, we have: 

 $$a_2 = a_1 \times r \Rightarrow 2 = a_1 \times r \quad (1)$$ 

 Given $a_5 = \frac{1}{4}$, we have: 

 $$a_5 = a_1 \times r^4 \Rightarrow \frac{1}{4} = a_1 \times r^4 \quad (2)$$ 

 Dividing equation (2) by equation (1): 

 $$\frac{\frac{1}{4}}{2} = \frac{a_1 \times r^4}{a_1 \times r} \Rightarrow \frac{1}{8} = r^3 \Rightarrow r = \sqrt[3]{\frac{1}{8}} = \frac{1}{2}$$ 

 Substituting $r = \frac{1}{2}$ into equation (1): 

 $$2 = a_1 \times \frac{1}{2} \Rightarrow a_1 = 4$$ 

 Thus, the geometric sequence is: 

 $$a_n = 4 \times \left(\frac{1}{2}\right)^{n-1} = 2^2 \times 2^{-n+1} = 2^{3-n}$$ | The reasoning uses the geometric sequence definition to set up equations for $a_2$ and $a_5$, solves for the ratio $r$ by dividing the equations, finds $r = 1/2$, then solves for $a_1 = 4$, and derives the general formula $a_n = 2^{3-n}$. 
 **YES** 
 The reasoning is rigorous and accurate. It correctly uses the definition of a geometric sequence, sets up equations based on the given terms ($a_2 = 2$ and $a_5 = 1/4$), and solves for the common ratio $r$ by dividing the equations. The calculation $\frac{\frac{1}{4}}{2} = \frac{1}{8} = r^3$ is correct, leading to $r = \frac{1}{2}$. Then, it substitutes back to find $a_1 = 4$. Finally, it expresses the general term $a_n$ in both exponential and simplified form ($2^{3-n}$). The steps are logical, clear, and mathematically sound, with no errors or gaps. This is a high-quality analysis. |

