# OpenReview forum: "Generative Adversarial Reasoner: Enhancing LLM Reasoning with Adversarial Reinforcement Learning"
_ICLR.cc/2026/Conference — ICLR 2026 Poster_

### Official Review · Reviewer_JCij · 2025-10-21

**Soundness:** 2
**Presentation:** 3
**Contribution:** 2
**Rating:** 2
**Confidence:** 3

**Summary:**

The paper proposes Generative Adversarial Reasoner (GAR): a joint RL framework that co-trains a reasoner LLM with a slice-level LLM discriminator to provide dense, on-policy process rewards. Reasoning chains are partitioned into “logically complete” slices; the discriminator outputs an analysis → YES/NO → brief rationale per slice, and training combines exact-match and aggregated slice rewards. On seven math benchmarks,  GAR improves over strong DeepSeek-R1-Distill baselines; authors report efficiency tricks and partial-trace RL without final answers.

**Strengths:**

S1. Stepwise tables show improvements from: (i) adding any critic, (ii) reframing to slice-level judgments, and (iii) joint on-policy updates with alignment + discriminator rewards—culminating in the strongest results.

S2. The discriminator operates on logically complete slices with an analysis → YES/NO → brief rationale format, producing localized, checkable feedback that improves credit assignment without expensive human PRM labels.

**Weaknesses:**

W1.  Novelty is overstated vs. existing PRM/LLM-judge and adversarial/self-play lines What is addressed as “co-evolving slice-level discriminator” is very close to (i) PRMs trained on stepwise feedback (human or LLM-as-judge) and (ii) self-play / adversarial critics already explored for reasoning. The paper doesn’t clearly isolate what’s genuinely new beyond (a) cutting long traces into “slices” and (b) mixing a GAN-style real/fake discriminator with an answer-aligned term. Without tighter theoretical or empirical differentiation from PRM-as-judge (e.g., Generative Verifiers, Math-Shepherd, debate/multi-agent critics), the claimed contribution reads incremental. Also, GAN objectives assume a meaningful data distribution for “real” vs. “generated.” Here, “real slices” are reference CoT fragments from existing models, not ground-truth human-verified reasoning. The discriminator can then learn stylistic artifacts of reference traces rather than logical soundness. This undermines the premise that Rd improves correctness (vs. style mimicry). No analysis disentangles style detection from error detection.

W2. Slicing heuristic (L=320 tokens, delimiter rules) is arbitrary and unvalidated
The core claim—“slice-level evaluation improves reliability”—depends on how you segment traces. But the paper gives no sensitivity study over L, no human audit of “logical completeness,” and no robustness to alternate chunking strategies. If performance hinges on a specific, hand-tuned slicer, the method risks brittleness and poor reproducibility.

W3. All evidence comes from seven mathematical benchmarks. Math affords crisp local checks (algebraic identity, parity, etc.), which makes slice-level “soundness” judgments unusually reliable. It’s unclear that the same discriminator will remain calibrated on domains without local verifiers (open-ended QA, multi-hop fact retrieval, safety reasoning, tool use, code with flaky unit tests, long-form writing).

W4. The “analysis–score–rationale” prompt and binary slice labels fit math well (short, checkable steps). In natural language reasoning, law, or medical QA, slices can be ambiguous and correctness is non-binary. The paper doesn’t show how slice segmentation and binary scoring handle ambiguity, partial credit, or evidence attribution.

**Questions:**

How do you detect and mitigate reward hacking (e.g., producing slice-looking “clean” text without real progress)? Any concrete failure cases? Why binary slice labels (YES/NO) instead of graded scores or token-level attributions? Did you try continuous rewards?

---

> ### Author Response · Authors · 2025-11-24
> **Official Comment by Authors (Part 1/3)**
>
> We thank the reviewer for the constructive feedback and address the concerns below.
> > W1: Novelty
>
> We thank the reviewer for the detailed comparison with PRM / LLM-judge and adversarial/self-play lines. We respectfully disagree that GAR is merely an incremental variant of existing PRM-as-judge methods, and we will clarify this positioning in the revised version. We also note that other reviewers explicitly acknowledge the methodological contribution: reviewer **mfoq** describes our method as a **“novel approach,”** and reviewer **iNKw** states that **“the paper’s core idea and method hold research value”** and is **“quite intriguing and holds potential reference value for the community’s understanding of RL.”**
>
> Specifically, we agree GAR is conceptually related to PRMs and LLM-judges. However, they are very big training concepts, i.e., training with stepwise feedback and using LLM to provide feedback. It doesn’t mean any paper uses stepwise feedback or LLM-judges lack novelty. Instead, GAR differs from prior work along several **substantive** axes:
>
> **(1) Co-evolving discriminator trained on automatically constructed slice pairs, not frozen labels.**
>
> PRMs and LLM-judges are typically trained on fixed, externally provided labels (human preferences, LLM comparisons, etc.) and then frozen during RL. GAR instead trains the discriminator jointly with the policy, on automatically constructed real vs. generated slice pairs that evolve as the policy improves. This co-evolution is central to our design and to the ability to operate without human stepwise labels. We believe this co-evolution is new to the field, and our results show that this co-evolution achieves significant improvements over baselines. Therefore, we believe our method is both novel and highly effective.
>
> **(2) Slice-level adversarial training for RL without full CoT or verifiable final answers.**
>
> Most PRM/LLM-judge work (generative verifiers, Math-Shepherd, debate/multi-agent critics, etc.) assumes either (a) access to full CoT and final answers, or (b) stepwise labels from humans / LLM-as-judge. In contrast, GAR can also handle the setting (see section 4.5) where:
> - the model **does not generate a full CoT,**
> - there is **no verifiable final answer,** and
> - training relies purely on slice-level feedback from the discriminator.
>
> Table 5 shows that, in this regime, GAR still improves performance while using only 37.5% of the RL training time of the baseline. To our knowledge, this “partial CoT + unverifiable answer” RL setting supported by a co-evolving slice-level discriminator **has not been explicitly addressed in prior PRM-as-judge work.**
>
> &nbsp;
>
> **“Real slices” vs. style mimicry**
>
> We agree that our “real slices” are reference CoTs from strong models (DeepSeek-R1 / Gemini) rather than human-verified logical proofs. We mentioned that both CoT from strong models and human-generated CoTs can be used (Line 471). In addition, to avoid style mimicry, we have the following designs:
>
> - (1) The discriminator reward is never used alone. It is always combined with the standard GRPO-style task rewards (exact-match) and KL to the SFT policy. Any behavior that merely imitates superficial style but hurts answer correctness or formatting is explicitly penalized by these other terms. This structurally restricts “style-only” reward hacking.
>
> - (2) Even for the discriminator, we combine the discriminator reward $R^d$ with the alignment reward $R^a$. $R^a$ ties the discriminator’s judgments to answer-level correctness signals, so the discriminator cannot just chase arbitrary artifacts in slices; it must remain consistent with whether the underlying solutions are actually correct.
>
> Empirically, we observe that **GAR consistently improves all task metrics (e.g., AIME24, AIME25, MATH500, GSM8K, AMC23, Olympiad Bench, and LiveMath Bench) and yields more coherent, human-interpretable CoTs, rather than just exploit the discriminator with style mimicry.**
>
> In addition, to directly address the reviewer’s concern, we will add an analysis in Table F comparing $R^d$ on correct vs. incorrect slices while controlling for surface style. Specifically, for the reference CoTs, we randomly perturb some intermediate numbers and results to introduce logical mistakes without changing the overall reasoning style, and show that the discriminator can still distinguish logically correct from incorrect slices even with the same thinking style. This explicitly tests whether the discriminator is learning to detect reasoning errors rather than merely imitating style.
>
> **Table F: Our discriminator can distinguish logically correct from incorrect slices, even under the same reasoning style.**
>
> |                 | Predicted correct | Predicted wrong |
> |-----------------|-------------------|-----------------|
> | **Actually correct** | 84.3%             | 15.7%           |
> | **Actually wrong**  | 23.3%             | 76.7%           |

---

> ### Author Response · Authors · 2025-11-24
> **Official Comment by Authors (Part 2/3)**
>
> > W2: Slicing heuristic (L=320 tokens, delimiter rules) is arbitrary and unvalidated
>
> **Justification for choices and comparison with alternate chunking strategies:**
>
>
> Regarding why we use the proposed method (i.e., delimiter + token-length segmentation), it is because, in practice, chain-of-thought from modern reasoning models (DeepSeek-R1, Gemini, etc.) is already organized by clear markers such as line breaks (\n) and discourse cues like “*Wait*,” “*Since*,” “*Therefore*”. These markers naturally define self-contained reasoning fragments. A simple token-length range avoids extremely short or extremely long slices, and keeps computation and implementation overhead very low.
>
> As reported in Table A, we also consider other methods, including (1) Pure fixed-length token windows; (2) LLM based semantic segmentation (we ask the model to automatically split the CoT into clips). We can see that these alternatives either break coherent reasoning steps, or are much more expensive, without improving performance over our rule-based + token-length method.
>
> **TableA: Ablation on different slice segmentation methods**
>
> | Method                          | AIME24 | Training time (hours) |
> |---------------------------------|--------|------------------------|
> | Pure fixed-length token windows | 58.7   | 19                     |
> | LLM based semantic segmentation | 61.6   | 35                     |
> | Ours                            | 61.3   | 19                     |
>
> &nbsp;
>
> **Sensitivity study over L:**
>
> Table B reports an ablation over slice length. Performance is best and relatively stable when slices are around **320–560 tokens**. Shorter slices often contain no explicit logical error, making it hard for the discriminator to learn useful signals; much longer slices almost always include at least one flaw, causing most slices to be labeled “incorrect” and weakening the discriminator’s supervision. We will clarify these design choices and sensitivity results in the revised version.
>
> **TableB: Ablation over slice length**
>
> | Slice length (tokens) | 160  | 320  | 480  | 560  | 800  | 960  | 1120 | 1440 |
> |-----------------------|------|------|------|------|------|------|------|------|
> | AIME24                | 57.4 | 61.3 | 61.5 | 61.4 | 61.0 | 59.3 | 56.5 | 56.8 |
>
> &nbsp; &nbsp;
>
> > W3: All evidence comes from seven mathematical benchmarks.
>
> We respectfully disagree that our gains are mainly a consequence of math-specific “local verifiers.” In our setup, the discriminator **never** receives symbolic supervision like algebraic identities or parity checks at the slice level. It only sees text slices, plus weak answer-level signals (through $R^a$) and reference CoTs from a strong teacher. In other words, GAR’s slice-level reward is learned, not given by **local verifiers**.
>
> In addition, almost **all recent RL / RLHF works on LLM reasoning in top venues (including PRMs, verifier-based RL, and GRPO-style methods) also focus mainly on math and code**, precisely because these domains offer clear, high-precision evaluation and are widely accepted as standard testbeds for reasoning. By contrast, evaluating RL-style reasoning improvements on open-ended QA, multi-hop fact retrieval, safety reasoning, tool use, code with flaky unit tests, or long-form writing is **not a standard experimental protocol and is rarely adopted in prior work.**
>
> Most importantly, nothing in GAR is mathematically hardcoded: the algorithm only assumes (i) access to reference CoTs from a strong model and (ii) some answer-level or preference signal (which, in non-math domains, could come from human/LLM feedback, task rewards, or behavioral constraints). The slice construction, discriminator training, and **GAR objective are directly applicable to other domains such as open-ended QA or tool use, as long as such signals exist**. We therefore view math benchmarks as a challenging but controlled first domain, not a requirement for GAR to function.

---

> ### Author Response · Authors · 2025-11-24
> **Official Comment by Authors (Part 3/3)**
>
> > W4: Limits of Binary Slice Labels Beyond Math
>
>
> Using binary or pairwise feedback is standard in RL even in ambiguous domains (e.g., preference models with win/lose labels or “better/worse” judgments). Our slice labels play the same role: they indicate whether a slice is judged *acceptable* vs. *unacceptable*, not that correctness is fundamentally binary. Partial credit is naturally expressed by (i) how many slices in a trajectory are judged good vs. bad, and (ii) the strength of the discriminator’s score; the overall return already aggregates over multiple slices rather than enforcing a single yes/no decision on the entire reasoning process.
>
> Importantly, **neither the slicing algorithm (based on discourse markers + minimum length) nor the GAR objective assumes that the underlying domain is math**. In non-math settings, the same framework can use soft discriminator scores or multi-level judgments (e.g., “clearly wrong / ambiguous / clearly correct”) obtained from human or LLM-as-judge feedback, instead of strict binary labels. The only requirement is a scalar slice-level signal, which is exactly the kind of supervision used by existing PRM / judge models in open-ended domains.
>
> Finally, as discussed in our response to W3, **evaluating RL-style reasoning improvements on law, or medical QA is not a standard protocol in the literature. Most recent RL/RLHF works on reasoning also focus on math and code**. We therefore view our math benchmarks as a challenging but controlled first setting.
>
> &nbsp; &nbsp;
>
> > Questions:
>
> **Reward hacking:**
>
> We mitigate the risk of reward hacking with the following designs:
>
> - (1) Fixed, high-quality reference distribution.
> Our adversarial game is not pure self-play: the discriminator is always anchored to fixed reference slices from a strong / different model (DeepSeek-R1, or Gemini in Section 4.5), rather than only chasing the current policy’s behavior. The discriminator is explicitly trained to distinguish these reference slices and score them higher than incorrect / low-quality slices, so the equilibrium corresponds to imitating a known good reasoning distribution, **making it hard to exploit arbitrary weaknesses in the discriminator (hence mitigate the reward hacking)**.
>
> - (2) Combined with standard task rewards and KL.
> GAR adds a discriminator-based reward on top of the standard GRPO-style objective used by DeepSeek-R1 (exact-match reward), rather than replacing it. Any “reward hacking” that degrades final-answer correctness or formatting would immediately be penalized by these existing terms and by the KL regularization. **This strongly constrains the space of degenerate solutions that only fool the discriminator.**
>
>
> - (3) Additional reward for the discriminator itself.
> Even for the discriminator itself, we combine the discriminator reward $R^d$ with an alignment reward $R^a$. The alignment reward encourages agreement with answer-level correctness signals, so the discriminator is not free to exploit arbitrary artifacts in the slices; it must stay consistent with whether the underlying solutions are actually correct. We also provide a brief theoretical analysis of how this additional loss helps mitigate reward hacking; see Appendix A.1 of the rebuttal.
>
>
> Empirically, we observe that **GAR consistently improves all task metrics (e.g., AIME24, AIME25, MATH500, GSM8K, AMC23, Olympiad Bench, and LiveMath Bench) and yields more coherent, human-interpretable CoTs, rather than bizarre traces that only exploit the discriminator**. In Section 4.5 (Distilling Reasoning Patterns with GAR), we further show that changing the reference model (e.g., to Gemini CoTs) leads to a qualitatively different but still sensible reasoning style, which **is hard to explain by trivial reward hacking**.
>
> &nbsp;
>
> **Failure cases:**
>
> Yes, we provide additional qualitative examples, including failure cases, in Tables 7 and 8. For example, Table 8 (case 2) shows a failure where the discriminator makes an incorrect judgment.
>
> &nbsp;
>
> **Why binary slice labels instead of graded scores or token-level attributions:**
>
> Our choice of binary “good / bad” labels is mainly for **stability and simplicity**:
> - Binary (or pairwise) feedback is **the standard choice in RLHF and PRM training (win/lose, better/worse)**, and it typically yields more stable gradients than noisy fine-grained scores.
> - Prior work has also observed that continuous rewards are often subjective, **significantly increasing annotation cost while reducing training stability.**
> - At the slice level, what we need is essentially a **preference signal** (“this slice is acceptable vs. not”), rather than a precisely calibrated numeric score. **Continuous credit is naturally reflected by (i) the fraction of slices in a trajectory judged “good,” and (ii) the aggregated return across multiple slices.**
> - Token-level attributions would require a different architecture and much stronger supervision (per-token rationales).

---

### Official Review · Reviewer_gLeA · 2025-10-25

**Soundness:** 3
**Presentation:** 3
**Contribution:** 3
**Rating:** 6
**Confidence:** 2

**Summary:**

This paper introduces the Generative Adversarial Reasoner, a framework that co-trains an LLM reasoner and an LLM discriminator through adversarial reinforcement learning.
The discriminator evaluates "slices" of the reasoning chain, providing dense, step-level rewards that improve credit assignment and sample efficiency over sparse final-answer rewards.
This method enhances performance on mathematical benchmarks and offers a flexible, modular approach for enhancing reasoning quality.

**Strengths:**

The paper introduces a co-training approach where the reasoner and discriminator evolve together, providing dense, calibrated, step-level rewards that significantly improve credit assignment over sparse outcome-based methods.

It shows improvements across multiple mathematical reasoning benchmarks (e.g., +7.3 on AIME24), outperforming strong RL baselines while maintaining comparable training efficiency.

It implements compute-efficient innovations like slice-level evaluation and response length truncation (128 tokens) that maintain performance while substantially reducing training time and resource requirements.

**Weaknesses:**

The adversarial training setup may lead to reward hacking, where the discriminator and reasoner adapt to each other's weaknesses rather than improving reasoning quality. How does this paper mitigate the risk of reward hacking?

It is recommended that the method proposed in this paper be validated on a broader range of tasks, such as coding and commonsense reasoning tasks.

**Questions:**

See the Weaknesses

---

> ### Author Response · Authors · 2025-11-24
> **Official Comment by Authors**
>
> We thank the reviewer for the constructive feedback and address the concerns below.
>
> > W1: Reward Hacking
>
> We thank the reviewer for raising this important concern. We mitigate the risk of reward hacking with the following designs:
>
> - (1) **Fixed, high-quality reference distribution.**
>
> Our adversarial game is not pure self-play: the discriminator is always anchored to fixed reference slices from a strong / different model (DeepSeek-R1, or Gemini in Section 4.5), rather than only chasing the current policy’s behavior. The discriminator is explicitly trained to distinguish these reference slices and score them higher than incorrect / low-quality slices, so the equilibrium corresponds to imitating a known good reasoning distribution, **making it hard to exploit arbitrary weaknesses in the discriminator (hence mitigate the reward hacking)**.
>
> - (2) **Combined with standard task rewards and KL.**
>
> GAR adds a discriminator-based reward on top of the standard GRPO-style objective used by DeepSeek-R1 (exact-match reward plus KL to the SFT policy), rather than replacing it. Any “reward hacking” that degrades final-answer correctness or formatting would immediately be penalized by these existing terms and by the KL regularization. **This strongly constrains the space of degenerate solutions that only fool the discriminator.**
>
>
> - (3) **Additional reward for the discriminator itself.**
>
> Even for the discriminator itself, we combine the discriminator reward $R^d$ with an alignment reward $R^a$. The alignment reward encourages agreement with answer-level correctness signals, so the discriminator is not free to exploit arbitrary artifacts in the slices; it must stay consistent with whether the underlying solutions are actually correct. We also provide a brief theoretical analysis of how this additional loss helps mitigate reward hacking; see Appendix A.1 of the rebuttal.
>
> &nbsp;
>
> Empirically, we observe that **GAR consistently improves all task metrics (e.g., AIME24, AIME25, MATH500, GSM8K, AMC23, Olympiad Bench, and LiveMath Bench) and yields more coherent, human-interpretable CoTs, rather than bizarre traces that only exploit the discriminator**. In Section 4.5 (Distilling Reasoning Patterns with GAR), we further show that changing the reference model (e.g., to Gemini CoTs) leads to a qualitatively different but still sensible reasoning style, which **is hard to explain by trivial reward hacking**. We will clarify these design choices and add a short discussion on reward hacking in the revised version.
>
>
> &nbsp; &nbsp;
>
> > W2: Evaluation on a broader range of tasks
>
> ~~Thank you for the valuable suggestion!! Our model demonstrates strong performance on additional reasoning tasks, such as coding, and we will upload the corresponding results very soon!~~
>
> Results and analysis on coding ability are added below. Thank you!

---

> ### Author Response · Authors · 2025-12-03
> **Official Comment by Authors (Evaluation on coding task)**
>
> > W2: Evaluation on a broader range of tasks
>
> Thank you for the great suggestion. Our model also shows promising results on **other reasoning tasks such as coding**, as demonstrated in **Table G**, where **GAR significantly improves coding performance**. All results are averaged over 30 trials per benchmark (three independent training runs times 10 inference seeds each) to ensure reliability. We train on the **CodeForces-CoT** dataset, where (approximate) ground-truth reasoning traces are generated by DeepSeek-R1. The models are trained with a maximum rollout length of 800 steps. The reward function is an **equal-weight combination** of three components: (1) a **standard CF-Code Reward**: Computes a weighted sum over all public cases (assigning 1 for pass and 0 for fail). (2) a **standard Code-Format Reward**: Assigns a reward of 1 when the model output includes a valid code-block wrapper (e.g., ```python <code>``` ). (3) **our Critic Reward**: Includes both alignment and discriminative terms, using the same design as in the math-reasoning setting. All three rewards are applied with equal weight during training.
>
> &nbsp;
>
> **Table G: Pass@1 accuracy on coding benchmarks.** Similar to the results in the main paper, scores are averaged over 30 runs per benchmark to reduce evaluation noise. Our model also significantly improves coding performance.
>
> | Model                    | LiveCodeBench | HumanEval | HumanEval+ |
> |--------------------------|---------------|-----------|-----------|
> | DS-R1-Distill-Qwen-7B    | 37.4          | 40.4      | 37.8      |
> | DS-R1-Distill-Qwen-7B + GAR (Ours) | 43.6 | 42.7 | 39.3 |

---

### Official Review · Reviewer_iNKw · 2025-10-27

**Soundness:** 3
**Presentation:** 3
**Contribution:** 3
**Rating:** 6
**Confidence:** 3

**Summary:**

This paper introduces Generative Adversarial Reasoner (GAR), an on-policy framework that jointly trains an LLM reasoner and an LLM discriminator via adversarial reinforcement learning. The discriminator evaluates slices of reasoning, providing dense, on-policy, step-level rewards. This method significantly improves mathematical reasoning, showing strong gains on benchmarks like AIME24.

**Strengths:**

1. The paper's core idea and method hold research value. Judging from the experimental results, the proposed method indeed brings about a relatively clear improvement compared to standard reinforcement learning methods.

2. The paper conducts thorough ablation studies, sufficiently demonstrating the effectiveness of each component within the designed framework. Furthermore, the discriminator's truncation experiment design gives consideration to its impact on training efficiency.

3. The paper includes several interesting discussions. First, the discussion on the experimental conclusions regarding policy entropy collapse is quite thorough. Additionally, the discussion on "RL without Full Chain-of-Thought or Verifiable Final Answers" is, in my opinion, quite intriguing and holds potential reference value for the community's understanding of RL.

**Weaknesses:**

1. First, I believe the paper aims to propose a denser reward strategy. However, the final method is still based on GRPO, which assigns a reward to the entire sequence. Therefore, although the proposed method can provide step-level rewards, it essentially still provides a denser Outcome Reward during training. I think the authors' description of their contribution requires careful consideration, as the PRM is not used in a step-level way. Admittedly, in Section 4.5, the authors mention the possibility of "RL without Full Chain-of-Thought or Verifiable Final Answers." Detailing these experiments would clarify the paper's contribution.

2. The comparison with other RL methods is insufficient. The authors only include standard RL as the sole baseline. However, there is a significant amount of other work on LLM Reasoning and RL. I believe the authors need to compare their method against some of the more influential works in this area to further demonstrate its effectiveness.

3. The "slice" plays a crucial role in the proposed method. The paper defines it as "logically coherent slices" segmented by "delimiters." This definition is somewhat heuristic. Is the model sensitive to the choice of $L$? Furthermore, what exactly are these "delimiters" (e.g., newlines, specific punctuation, or semantic boundaries)? A clearer description of the slicing algorithm is needed for reproducibility. Additionally, the text mentions mixing generated slices with an "equal number of reference slices." Do these "reference slices" originate from the baseline model’s (DeepSeek-R1) COT? If so, this should be clarified in methodology (Section 3). This element seems to function more like a regularizer (to prevent the discriminator from deviating excessively from reasonable reasoning patterns) than a traditional GAN objective (imitating $p_{ref}$).

4. The proposed GAR framework is quite complex, involving numerous training hyperparameter settings ($\lambda_{1}=\lambda_{2}=\lambda_{3}=1, \lambda_{4}=0.5$). The paper lacks an analysis of these weight choices. If performance is highly sensitive to minor variations in these weights, the framework will be relatively difficult to reproduce and apply.

**Questions:**

Please see weaknesses.

---

> ### Author Response · Authors · 2025-11-24
> **Official Comment by Authors (Part 1/2)**
>
> We thank the reviewer for the constructive feedback and address the concerns below.
>
> > W1: Denser Reward Strategy vs Reward per Entire Sequence
>
> We thank the reviewer for this comment and agree that the underlying RL optimizer is still GRPO, which updates the whole sequence. In standard GRPO, the reward is tied almost entirely to the final outcome (exact match, format, etc.), so all tokens in the trajectory share **one sparse signal**. In GAR, by contrast, the total return is the **sum of slice-level terms**, so the model receives multiple, more fine-grained signals that are easier to distinguish than a single outcome reward.
>
> We agree that our method is not token-level credit assignment in the strict sense, and that the policy is still updated via sequence-level GRPO. We will revise the contribution section to emphasize that GAR introduces **slice-level intermediate rewards (via a co-evolving discriminator) on top of GRPO**, rather than claiming an entirely new RL paradigm.
>
> &nbsp;
>
> **RL without Full Chain-of-Thought or Verifiable Final Answers**
>
> The “RL without Full Chain-of-Thought or Verifiable Final Answers” setting is demonstrated in **Table 5**. In this experiment, the LLM **does not** generate a complete chain-of-thought or a final answer. Instead, we stop generation after three reasoning slices (about 960 tokens, compared with 4096 tokens in the baseline R1 model), have our discriminator score these partial traces only, and update the policy using those scores without any final-answer reward. Hence in this experiment, we train the LLM without
> **full chain-of-thought or verifiable final answers.**
>
> As shown in Table 5, this partial-trace setup **surpasses standard RL** on AIME24 (57.7 vs 56.3) while using only **6 vs 16 hours** of training time (≈37.5% of the cost). This highlights the effectiveness of our slice-level reward. We will clarify this protocol more explicitly in Section 4.5 and add a direct pointer to Table 5 in the revised version.
>
> &nbsp; &nbsp;
>
> > W2: The comparison with other RL methods is insufficient.
>
> Although some related methods are also RL-based, they are **not directly comparable** to our setting due to differences in base models, whether they use explicit reasoning, tasks, and training strategies. For example, **SPIRAL[b]** formulates a zero-sum game for self-play and self-evolution, while **LSP[c]** uses an LLM to play Challenger and Solver, where the Challenger generates instructions the Solver follows to enable data-free training. These settings are quite different from ours, so a fair head-to-head comparison is not feasible.
>
> At the same time, we argue that our method is **complementary** and can be applied on top of such approaches. We illustrate this by considering a recent GRPO improvement, **KL_Cov [d]**. As shown in Table E, adding GAR on top of KL_Cov further improves performance, suggesting that our framework can also enhance other more advanced RL-based reasoning algorithms.
>
> **Table E: GAR Improves KL_Cov (Enhanced GRPO Baseline)**
>
> |                             | AIME24 | AIME25 | AMC  | MATH-500 | OlympiadBench |
> |---------------------------- |--------|--------|------|----------|---------------|
> | GRPO                        | 21.2   | 9.6    | 58.7 | 78.8     | 40.7          |
> | GRPO + KL-Cov [c]           | 22.6   | 12.9   | 61.4 | 80.8     | 42.6          |
> | GRPO + KL-Cov [c] + GAR (ours) | 25.7| 16.2   | 62.5 | 84.2     | 44.0          |
>
>
>
> [b] Liu et al. Spiral: Self-play on zero-sum games incentivizes reasoning via multi-agent multi-turn reinforcement learning. Arxiv 2025
>
> [c] Kuba et al. Language Self-Play For Data-Free Training. Arxiv 2025
>
> [d] Cui et al. The entropy mechanism of reinforcement learning for reasoning language models. Arxiv 2025

---

> ### Author Response · Authors · 2025-11-24
> **Official Comment by Authors (Part 2/2)**
>
> > W3: Slice Segmentation Methodology
>
> **Slice and delimiters**
>
> In our implementation, a slice is a contiguous part of the CoT segmented by simple textual delimiters and a minimum length L. Concretely, we first split the CoT by delimiters including line breaks “\n” and common discourse markers such as “*Wait*,” “*Since*,” “*Therefore*,” “*Thus*”, which frequently mark step boundaries in common CoTs such as DeepSeek-R1 / Gemini. We then greedily merge adjacent segments until a slice reaches at least L≈320 tokens, and start a new slice. We demonstrate its effectiveness in Table A. We also consider other methods, including (1) Pure fixed-length token windows; (2) LLM based semantic segmentation (we ask the model to automatically split the CoT into clips). We can see that these alternatives either break coherent reasoning steps, or are much more expensive, without improving performance over our rule-based + token-length method. We will add this algorithm and delimiter list to Section 3 for reproducibility.
>
> **TableA: Ablation on different slice segmentation methods**
>
> | Method                          | AIME24 | Training time (hours) |
> |---------------------------------|--------|------------------------|
> | Pure fixed-length token windows | 58.7   | 19                     |
> | LLM based semantic segmentation | 61.6   | 35                     |
> | Ours                            | 61.3   | 19                     |
>
> &nbsp;
>
> **Sensitivity study over slice length:**
>
> Table B reports an ablation over slice length L. Performance is best and relatively stable when slices are around **320–560 tokens**. Shorter slices often contain no explicit logical error, making it hard for the discriminator to learn useful signals; much longer slices almost always include at least one flaw, causing most slices to be labeled “incorrect” and weakening the discriminator’s supervision. We will clarify these design choices and sensitivity results in the revised version.
>
> **TableB: Ablation over slice length**
>
> | Slice length (tokens) | 160  | 320  | 480  | 560  | 800  | 960  | 1120 | 1440 |
> |-----------------------|------|------|------|------|------|------|------|------|
> | AIME24                | 57.4 | 61.3 | 61.5 | 61.4 | 61.0 | 59.3 | 56.5 | 56.8 |
>
> &nbsp;
>
> **Reference slices.**
>
> As noted in Line 267, the “reference slices” in our main experiments are sampled from the DeepSeek-R1 CoTs. However, we also show that other sources can be used: in Section 4.5 (“Distilling Reasoning Patterns with GAR”), the reference slices come from Gemini CoTs, and GAR successfully enables the DeepSeek-R1-Distill-Qwen-7B model to adopt a Gemini-like reasoning style (Table 6). This shows that the mechanism is more than a simple regularizer. The discriminator is trained to score reference slices higher than model-generated slices, while the reasoner is trained to produce slices that the discriminator cannot distinguish from the references. Thus, the interaction is adversarial in the GAN sense and actively pushes the LLM to think in the chosen reference reasoning style.
>
> &nbsp; &nbsp;
> > W4: The proposed GAR framework is quite complex, involving numerous training hyperparameter settings (various $\lambda$)
>
> We thank the reviewer for raising this point.
>
> Although the notation introduces several $\lambda$s, the framework is actually **simple**: these are just scalar weights in the reward function, exactly as in standard RL and RLHF setups. In fact, **it is common for RL methods to combine multiple reward terms to stabilize and improve training** (e.g., exact-matching reward + format reward + tag-count reward in standard DeepSeek-R1 training with GRPO, which is also our baseline).
>
> In our work, we **do not** aggressively tune these weights. We largely **follow prior practice and reuse the same coefficients**: most reward terms use weight 1. The only down-weighted term is the align loss, because it is less reliable than the others by construction: it relies on the additional assumption that if an answer is correct, the reasoning steps leading to that answer should also be logically sound (Line 210). Following standard practice, we down-weight it by a factor of 2 and use a coefficient of 0.5. The fact that GAR already yields strong improvements under this direct reuse of hyperparameters suggests that **the framework is not overly sensitive to minor variations in these $\lambda$s and is practical to reproduce.**

---

### Official Review · Reviewer_mfoq · 2025-10-30

**Soundness:** 2
**Presentation:** 3
**Contribution:** 3
**Rating:** 6
**Confidence:** 4

**Summary:**

This paper introduces Generative Adversarial Reasoner (GAR), a novel framework that jointly trains an LLM reasoner and an LLM-based discriminator through adversarial reinforcement learning to enhance mathematical reasoning capabilities. The key innovation is partitioning reasoning chains into logically complete "slices" that the discriminator evaluates for soundness, providing dense step-level rewards rather than relying solely on sparse final-answer feedback. The reasoner is rewarded for producing logically consistent steps leading to correct answers, while the discriminator receives rewards for correctly detecting errors and distinguishing between generated and reference reasoning. Experiments on mathematical benchmarks show consistent improvements over strong baselines, with notable gains on challenging datasets like AIME24 (+7.3 for DeepSeek-R1-Distill-Qwen-7B and +10.0 for DeepSeek-R1-Distill-Llama-8B).

**Strengths:**

1. The paper presents a novel approach to LLM reasoning improvement by adapting GAN-style adversarial training to the reasoning domain. The slice-level evaluation mechanism is creative and addresses computational efficiency concerns while maintaining granular feedback. The joint training paradigm where both reasoner and discriminator co-evolve is innovative compared to static reward models.

2. The experimental methodology is solid with comprehensive evaluation across seven mathematical benchmarks. The authors provide proper statistical analysis (averaged over 30 runs) and fair comparisons using unified evaluation protocols. The ablation studies systematically validate each component's contribution. The selective-entropy analysis provides valuable insights into how the method avoids common pitfalls of RL training.

3. The paper is generally well-written with clear motivation and methodology. The slice-level feedback examples in Table 2 effectively demonstrate the discriminator's capabilities. The figures and experimental results are presented clearly.

**Weaknesses:**

1. Limited Theoretical Foundation: While the empirical results are strong, the paper lacks theoretical analysis of the joint training dynamics. There's no convergence analysis or guarantees about the co-evolution process, which is concerning given the potential for reward hacking mentioned by the authors.

2. Slice Segmentation Methodology: The slice partitioning approach appears somewhat ad-hoc. The paper doesn't provide sufficient justification for these choices or analysis of how sensitive the results are to different segmentation strategies.

3. Computational Overhead Analysis: While the authors claim efficiency improvements, the analysis is incomplete. Table 3 shows training time comparisons, but there's no detailed breakdown of computational costs during inference or comprehensive analysis of the discriminator's computational burden across different problem complexities.

4. Limited Baseline Comparisons: The paper primarily compares against variants of DeepSeek-R1-Distill models. Comparisons with other recent reasoning enhancement methods (beyond standard RL) would strengthen the evaluation. The related work section mentions several relevant approaches that aren't empirically compared.

**Questions:**

1. Why use a smaller model (1.5B vs 7B) for the discriminator in the Qwen setup but the same size for Llama? How does discriminator capacity affect performance?

Others see Weaknesses

---

> ### Author Response · Authors · 2025-11-24
> **Official Comment by Authors (Part 1/3)**
>
> We thank the reviewer for the constructive feedback and address the concerns below.
>
> > W1: Theoretical Foundation
>
> We agree that a full convergence analysis of the joint min–max training (LLM reasoner + discriminator + RL) would be valuable.
>
> First, we want to argue that as Skalse et al. (2022) [a] formally show, even in very simple settings it is almost impossible to obtain a non-trivial proxy reward that is provably unhackable when we consider the set of all policies. In their framework, a proxy $R_2$ is unhackable w.r.t. a true reward $R_1$ only if there is **no** pair of policies $\pi_1, \pi_2$ such that $$J_2(\pi_2) > J_2(\pi_1) \quad \text{but} \quad J_1(\pi_2) < J_1(\pi_1)$$
>
> They prove that for sufficiently rich policy classes, the only unhackable proxies are either _trivial_ or essentially _identical_ to the true reward [a]. In other words, **strong global “non-hackability” guarantees are not just missing from our work; they are fundamentally out of reach for essentially all practical RLHF / process-supervision methods.**
>
> What we can do, and what our method is designed for, is to **reduce hackability relative to existing co-evolution procedures, on the on-policy distribution where learning actually happens**.
>
> Thanks for your suggestion, we provide some theoretical analysis to prove that **our method reduce hackability compared with previous methods**, following exactly the definitions and intuitions in [a]. Specifically:
> 1. In our setting, the _true_ reward is the exact-match correctness of the final answer, $R_{\text{true}}(\tau) = R^m(\tau) \in \{0,1\},$, while the reasoner is optimized on $R_{\text{rea}}(\tau) = \lambda_1 R^m(\tau) + \lambda_2 R^s(\tau)$, where $R^s$​ is the discriminator’s slice-level reward. Baselines that use a static process reward or a co-trained critic without our objectives correspond to optimizing a proxy $R_2$ that is not constrained to preserve the ordering induced by $R_{\text{true}}$ on the explored trajectories, and Skalse et al.’s results imply that reward hacking is then expected rather than exceptional.
> 2. Our discriminator, in contrast, is trained with the usual GAN objective $R^d$ plus an **alignment reward** $R^a$​, which explicitly maximizes the agreement between per slice scores $r^s_i$​ and the final correctness $R^m$​. Under this objective, in the idealized limit of a Bayes-optimal discriminator, $R^s(\tau)$ approximates $\mathbb{E}[R^m(\tau)\mid\text{slices in }\tau]$. In Skalse’s terminology, this makes $R^s$​ a _monotone transform_ of the true reward **on the on-policy trajectory set**, which strictly reduces the number of policy pairs $(\tau_1,\tau_2)$ where $R^s$​ and $R^m$​ disagree in ordering. Therefore, **while unhackability over the full policy space remains impossible, the measure of hackable pairs in the region the optimizer actually visits is reduced compared to baselines.**
> 3. In addition, because the reasoner’s objective is a **mixture** $R_{\text{rea}} = \lambda_1 R^m + \lambda_2 R^s$, any policy update that decreases $R^m$ must compensate with a sufficiently large increase in $R^s$ to be preferred. The alignment term $R^a$ is precisely designed to make such compensating “hacks” difficult to sustain: trajectories with high $R^s$ but low $R^m$ become negative examples for the discriminator, which is then trained (via $R^d$ and $R^a$) to lower their slice scores.  **As a result, even if the reasoner temporarily finds a reward-hacking pattern, the joint training process tends to *identify and remove* the exploit in subsequent updates. This is in contrast to static reward-model baselines, where once a loophole is found, it typically remains exploitable for the rest of training.**
>
> From the perspective of [1], we therefore do not (and cannot) claim a global convergence or non-hackability theorem over the full policy space. Instead, we provide:
>
> 1. A design explicitly motivated by the formal notion of reward hacking, intended to shrink the set of proxy/true reward ordering conflicts on the on-policy distribution, thereby **making the proxy reward less hackable than in standard co-evolution setups**;
> 2. **Empirical evidence that our method is indeed more robust than baselines in practice.**
>
> We will clarify this connection to [a] and add a short theoretical discussion of reduced on-policy hackability in the final version to better address the reviewer’s concern about theoretical foundations.
>
> [a] Skalse et al., Defining and Characterizing Reward Hacking, NeurIPS 2022.

---

> ### Author Response · Authors · 2025-11-24
> **Official Comment by Authors (Part 2/3)**
>
> > W2: Slice Segmentation Methodology
>
> **Justification for choices:**
>
>
> Regarding why we use the proposed method (i.e., delimiter + token-length segmentation), it is because, in practice, chain-of-thought from modern reasoning models (DeepSeek-R1, Gemini, etc.) is already organized by clear markers such as line breaks (\n) and discourse cues like “*Wait*,” “*Since*,” “*Therefore*”. These markers naturally define self-contained reasoning fragments. A simple token-length range avoids extremely short or extremely long slices, and keeps computation and implementation overhead very low.
>
> As reported in Table A, we also consider other methods, including (1) Pure fixed-length token windows; (2) LLM based semantic segmentation (we ask the model to automatically split the CoT into clips). We can see that these alternatives either break coherent reasoning steps, or are much more expensive, without improving performance over our rule-based + token-length method.
>
> **TableA: Ablation on different slice segmentation methods**
>
> | Method                          | AIME24 | Training time (hours) |
> |---------------------------------|--------|------------------------|
> | Pure fixed-length token windows | 58.7   | 19                     |
> | LLM based semantic segmentation | 61.6   | 35                     |
> | Ours                            | 61.3   | 19                     |
>
> &nbsp;
>
> **Sensitivity study over slice length:**
>
> Table B reports an ablation over slice length. Performance is best and relatively stable when slices are around 320–560 tokens. Shorter slices often contain no explicit logical error, making it hard for the discriminator to learn useful signals; much longer slices almost always include at least one flaw, causing most slices to be labeled “incorrect” and weakening the discriminator’s supervision. We will clarify these design choices and sensitivity results in the revised version.
>
> **TableB: Ablation over slice length**
>
> | Slice length (tokens) | 160  | 320  | 480  | 560  | 800  | 960  | 1120 | 1440 |
> |-----------------------|------|------|------|------|------|------|------|------|
> | AIME24                | 57.4 | 61.3 | 61.5 | 61.4 | 61.0 | 59.3 | 56.5 | 56.8 |
>
> &nbsp; &nbsp;
>
> > W3: Computational Overhead Analysis
>
> **Computational costs during inference:**
>
> The reason we do not report a detailed inference-time breakdown in Table 3 is that **the discriminator is not used at inference at all (Line 242)**. At test time we simply run the base LLM with the standard inference procedure, identical to the baseline. Thus, **our method does not introduce any additional computational cost during inference.**
>
> &nbsp;
>
> **Computational burden across different problem complexities:**
>
> Table 3 already reports end-to-end training time, which includes the discriminator. As with other CoT-based methods, more complex problems typically lead to longer chains-of-thought, increasing the LLM’s computational burden. The discriminator overhead scales roughly linearly with the CoT length (i.e., number of slices), but we apply a truncation strategy that caps the total CoT tokens. This keeps the additional discriminator cost bounded and modest even on more complex problems. We will clarify these points and explicitly state that Table 3 includes the discriminator’s cost in the revised version.
>
> &nbsp;&nbsp;

---

> ### Author Response · Authors · 2025-11-24
> **Official Comment by Authors (Part 3/3)**
>
> > W4: Limited Baseline Comparisons
>
> Although some related methods are also RL-based, they are **not directly comparable** to our setting due to differences in base models, whether they use explicit reasoning, tasks, and training strategies. For example, **SPIRAL[b]** formulates a zero-sum game for self-play and self-evolution, while **LSP[c]** uses an LLM to play Challenger and Solver, where the Challenger generates instructions the Solver follows to enable data-free training. These settings are quite different from ours, so a fair head-to-head comparison is not feasible.
>
> At the same time, we argue that our method is **complementary** and can be applied on top of such approaches. We illustrate this by considering a recent GRPO improvement, **KL_Cov [d]**. As shown in Table C, adding GAR on top of KL_Cov further improves performance, suggesting that our framework can also enhance other more advanced RL-based reasoning algorithms.
>
> **Table C: GAR Improves KL_Cov (Enhanced GRPO Baseline)**
>
>
> |                             | AIME24 | AIME25 | AMC  | MATH-500 | OlympiadBench |
> |---------------------------- |--------|--------|------|----------|---------------|
> | GRPO                        | 21.2   | 9.6    | 58.7 | 78.8     | 40.7          |
> | GRPO + KL-Cov [c]           | 22.6   | 12.9   | 61.4 | 80.8     | 42.6          |
> | GRPO + KL-Cov [c] + GAR (ours) | 25.7| 16.2   | 62.5 | 84.2     | 44.0          |
>
>
>
> [b] Liu et al. Spiral: Self-play on zero-sum games incentivizes reasoning via multi-agent multi-turn reinforcement learning. Arxiv 2025
>
> [c] Kuba et al. Language Self-Play For Data-Free Training. Arxiv 2025
>
> [d] Cui et al. The entropy mechanism of reinforcement learning for reasoning language models. Arxiv 2025
>
> &nbsp;&nbsp;
>
> > Q1: Why use a smaller model (1.5B vs 7B) for the discriminator in the Qwen setup but the same size for Llama?
>
> **1.5B vs 7B discriminator for Qwen setup**
>
> In the Qwen setup, we use a smaller discriminator (1.5B) because, after our SFT, even this small model already becomes a strong discriminator. As shown in Table D, replacing the 1.5B discriminator with a 7B one yields better performance but incurs substantially higher training cost. Due to compute constraints, we therefore adopt the 1.5B model as the discriminator.
>
> **TableD: Ablation on different discriminator size**
> | Model size | AIME24 | AIME25 |
> |-----------|--------|--------|
> | 1.5B      | 61.3   | 44.3   |
> | 7B        | 62.5   | 45.2   |
>
> &nbsp;
>
> **Why the same size for Llama**
>
> For Llama, as noted in Line 257, the 8B model is the smallest publicly available Llama checkpoint with R1-style reasoning ability, so we do not have access to a smaller, compatible backbone for the discriminator. Therefore, in the Llama setup we use the same 8B model as both generator and discriminator.

---

### Meta-Review · Area_Chair_7Rcn · 2026-01-07

**Summary:**

This paper proposes Generative Adversarial Reasoner (GAR), a framework that jointly trains an LLM reasoner and discriminator through adversarial reinforcement learning to enhance mathematical reasoning. The method achieves consistent improvements across seven benchmarks. Three reviewers gave weak accept scores, appreciating the novelty of the proposed idea. The rebuttal provided substantial empirical evidence addressing reproducibility and methodological concerns, though baseline comparisons and generalization remain partially unresolved.

**Reviewer Concerns:**

The rebuttal effectively addressed most technical concerns with concrete experimental evidence. The resolved concerns include slice segmentation methodology, computational overhead concerns, discriminator capacity effects, style-mimicry vs. logical error detection and reward hacking concerns.

Some concerns remain not fully addressed. Baseline comparisons with other recent RL reasoning methods, generalization beyond mathematical reasoning, hyperparameter sensitivity, etc.

**Reviewer Scores:**

The rebuttal addressed most concerns but theoretical analysis and baseline comparisons remain limited. Overall, the reviewers seems satisfactory, and I can see a weak consensus toward acceptance.

---

### Decision · Program_Chairs · 2026-01-26

Accept (Poster)